# Locality-Based Mini Batching for Graph Neural Networks

## Abstract

Training graph neural networks on large graphs is challenging since there is no clear way of how to extract mini batches from connected data. To solve this, previous methods have primarily relied on sampling. While this often leads to good convergence, it introduces significant overhead and requires expensive random data accesses. In this work we propose locality-based mini batching (LBMB), which circumvents sampling by using fixed mini batches based on node locality. LBMB first partitions the training/validation nodes into batches, and then selects the most important auxiliary nodes for each batch using local clustering. Thanks to precomputed batches and consecutive memory accesses, LBMB accelerates training by up to 20x per epoch compared to previous methods, and thus provides significantly better convergence per runtime. Moreover, it accelerates inference by up to 100x, at little to no cost of accuracy.

## 1 Introduction

Modern neural networks commonly use stochastic mini-batch training to leverage large datasets and accelerate convergence. This strategy becomes highly non-trivial for connected data, since creating mini batches requires selecting a meaningful subset from the dataset, despite its connectedness. Graph neural networks (GNNs) typically rely on sampling a set of nodes from the graph to resolve this issue. However, graph sampling requires non-contiguous memory accesses, which significantly slows down training for large datasets. This severely limits their applicability to real-world graphs, which often consist of millions or billions of nodes.

The main question behind the connected mini-batching problem is: How do we choose the best nodes for constructing the next mini batch? To answer this question, we introduce the concept of *primary* and *auxiliary* nodes. Primary nodes are those for which we compute a prediction *in this batch*, typically a set of training nodes. Auxiliary nodes only help with computing the primary nodes' outputs. This distinction allows us to provide a meaningful neighborhood for every node's prediction, while ignoring irrelevant parts of the graph. Note that primary nodes in one batch can be auxiliary nodes in another batch.

This distinction splits the main question into two parts: 1. How do we choose the primary nodes for a mini batch? 2. How do we choose the auxiliary nodes for a given set of primary nodes? Having split the problem like this, we see that most previous works actually only focus on the second question and just choose a uniformly random subset of nodes as primary nodes (Hamilton et al., 2017; Zou et al., 2019). However, the nodes in a graph are not independent, and a better approach can lead to large improvements in runtime.

In this work, we propose one simple answer for both questions: Locality. Choosing a set of locally connected nodes is advantageous both from a computational and a predictive perspective. It allows us to share computation between multiple nodes, keeps memory accesses local, and reduces the mini batch's memory footprint. The predictions of most GNNs already leverage the fact that nearby nodes are more important. Previous works have even shown that incorporating locality can improve GNN accuracy (Klicpera et al., 2019b; Huang et al., 2021). Batching nearby primary nodes together can create synergies since one primary node can leverage another one's auxiliary nodes.

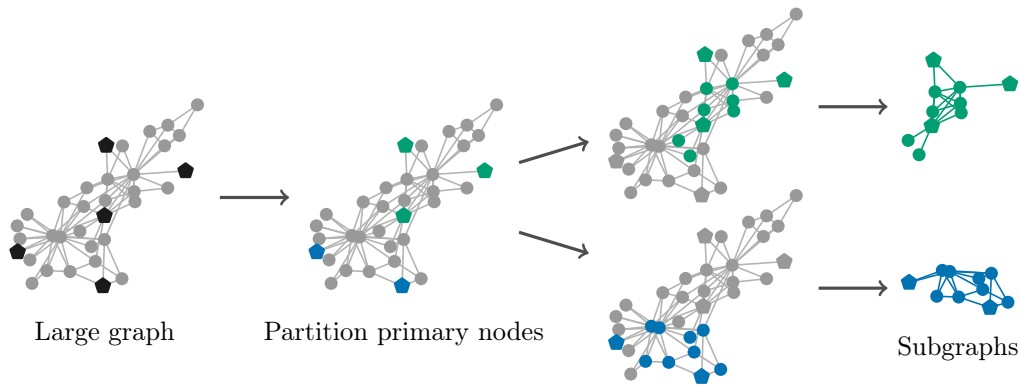

Large graph     Partition primary nodes     Subgraphs

Select auxiliary nodes

Figure 1: Locality-based mini batching. The primary (e.g. training) nodes are indicated by pentagons. These nodes are first partitioned into batches, e.g. using graph partitioning. We then use local clustering to select the auxiliary nodes of each batch, e.g. the neighbors with top-$k$ personalized PageRank (PPR) scores. Finally, we generate a batch using the induced subgraph of all selected nodes, but only calculate the outputs of the primary nodes we chose when partitioning. Batches can overlap and do not need to cover the whole graph.

More specifically, we propose to use either graph partitioning or node distances to select mini batches of primary nodes, and local clustering to select their auxiliary nodes. We then use the subgraph induced by these nodes as a mini batch. See Fig. 1 for an overview of this process. Importantly, these mini batches can be computed a priori, and loaded from a cache to ensure efficient memory access. During training, we counteract the effect of correlated mini batches with learning rate and batch scheduling. Overall, our method achieves an up to 20x improvement in time per epoch, with similar final accuracy. This faster time per epoch more than makes up for any slow-down in convergence per step. Its speed advantage grows even further for the common setting of low label ratios, since our method avoids computation on irrelevant parts of the graph. As opposed to most previous works, our method can successfully be used for both training *and* inference. It accelerates inference by up to 100x compared to previous methods that achieve similar accuracy. In summary, our core contributions are:

- Locality-based mini batching (LBMB): A general mini-batching method that works for a variety of GNNs and datasets. It substantially accelerates both training and inference without sacrificing accuracy, especially for small label ratios.
- We examine the impact of fixed, correlated mini batches on gradient estimation, and propose training methods to mitigate these effects.
- An extensive and fair experimental evaluation of scalable training methods, covering four datasets, two GNNs, and both training and inference.

## 2 BACKGROUND

**Graph neural networks.** We consider a graph $\mathcal{G} = (\mathcal{V}, \mathcal{E})$ with node set $\mathcal{V}$ and (possibly directed) edge set $\mathcal{E}$. $N = |\mathcal{V}|$ denotes the number of nodes, $E = |\mathcal{E}|$ the number of edges, and $\boldsymbol{A} \in \mathbb{R}^{N \times N}$ the adjacency matrix. GNNs use one embedding per node $\boldsymbol{h}_u \in \mathbb{R}^H$ and edge $\boldsymbol{e}_{(uv)} \in \mathbb{R}^{H_e}$, and update them in each layer via message passing between neighboring nodes. Most GNNs can be expressed via the following equations:

$$\boldsymbol{h}_u^{(l+1)} = f_{\text{node}}(\boldsymbol{h}_u^{(l)}, \underset{v \in \mathcal{N}_u}{\text{Agg}} [f_{\text{msg}}(\boldsymbol{h}_u^{(l)}, \boldsymbol{h}_v^{(l)}, \boldsymbol{e}_{(uv)}^{(l)})]), \tag{1}$$

$$\boldsymbol{e}_{(uv)}^{(l+1)} = f_{\text{edge}}(\boldsymbol{h}_u^{(l+1)}, \boldsymbol{h}_v^{(l+1)}, \boldsymbol{e}_{(uv)}^{(l)}). \tag{2}$$

The node and edge update functions $f_{\text{node}}$ and $f_{\text{edge}}$, and the message function $f_{\text{msg}}$ can be implemented using e.g. linear layers, multi-layer perceptrons (MLPs), and skip connections.

The node's neighborhood $\mathcal{N}_u$ is usually defined directly by the graph $\mathcal{G}$ (Kipf & Welling, 2017), but can be generalized to consider larger or even global neighborhoods (Klicpera et al., 2019b; Alon & Yahav, 2021), or feature similarity (Deng et al., 2020). The most common aggregation function Agg is summation, but mean, min, standard deviation, and other alternatives have also been explored (Corso et al., 2020; Geisler et al., 2020). Edge embeddings $\boldsymbol{e}_{(uv)}$ are not present in many GNNs, but some variants rely on them exclusively (Chen et al., 2019). GNNs commonly use node features $\boldsymbol{X} \in \mathbb{R}^{N \times F}$ for the input embeddings $\boldsymbol{h}_u^{(0)}$, but they can also be augmented with positional encodings (You et al., 2019; Dwivedi et al., 2020) or node IDs (Vignac et al., 2020). See App. A for related work in scalable GNNs.

## 3 Locality-based mini batching

An efficient GNN mini-batching method needs to consider both computational aspects and convergence per gradient step. Previous works have largely been focused on sampling methods that improve GNN convergence, and treated computational issues mostly as an afterthought. By focusing more on computational aspects like memory access times, we can accelerate training by multiple orders of magnitude, and therefore more than make up for any disadvantage in convergence per step.

If we skip details like caching the computational perspective is rather simple: Save as many operations and memory accesses as possible while maintaining a good GNN approximation. When the auxiliary nodes of different primary nodes in a batch are shared, we only have to compute their embeddings once and save computation and memory accesses. We therefore aim to group together primary nodes in a way that maximizes the number of shared auxiliary nodes. In locality-based mini batching (LBMB), we do this in two steps:

1. Obtain the $k$ most important auxiliary nodes for each primary node. Batch together primary nodes so that the union of their auxiliary nodes is smallest, yielding the primary node partition $P_{\mathrm{I}}$.
2. Select the most important auxiliary nodes $\mathcal{S}_{\mathrm{II}}$ for each subset of primary nodes $\mathcal{S}_{\mathrm{I}} \in P_{\mathrm{I}}$.

This process yields a single, fixed set of batches. We thus only need to perform it once during preprocessing. We then cache each mini batch in consecutive blocks of memory, thereby circumventing expensive random data accesses. This significantly accelerates training, allows efficient distributed training, and enables even expensive node selection procedures. In contrast, most previous methods select both primary and auxiliary nodes randomly in each epoch (Ying et al., 2018; Zeng et al., 2020), which incurs significant overhead. Our experiments show that our more efficient memory accesses clearly outweigh the slightly better gradient estimates gained from re-sampling in each epoch (see Sec. 5). We will next describe the details of LBMB's two main steps.

### 3.1 Primary node partitioning

**Optimal partitioning.** In this step we are interested in finding the partition $P_{\mathrm{I}}$ with the highest number of shared auxiliary nodes. Naïvely, we could find $P_{\mathrm{I}}$ by comparing the auxiliary node overlaps achieved by every possible partition. This is clearly intractable since the number of partitions increases exponentially with the number of primary nodes. An obvious way of accelerating this is a greedy approach, in which we iteratively put those nodes into a batch that has the largest overlap. Unfortunately, this would still require computing the overlap for every pair of primary nodes. This requires a quadratic runtime $\mathcal{O}(N^2)$, which is intractable for large datasets. We must therefore rely on heuristics to obtain a scalable, well-performing primary node partitioning algorithms.

**Distance-based partitioning.** We propose two methods that leverage graph locality as a heuristic for effective node partitioning. The first is based on node distances. In this approach we first compute the pairwise node distances between nodes that are close in the graph. A common node distance measure in undirected graphs are random walks with restart, or personalized PageRank (PPR) (Page et al., 1998). The PPR matrix is given by

$$\boldsymbol{\Pi}^{\mathrm{ppr}} = \alpha(\boldsymbol{I}_N - (1 - \alpha)\boldsymbol{D}^{-1}\boldsymbol{A})^{-1}, \tag{3}$$

with the teleport probability $\alpha \in (0, 1]$ and the diagonal degree matrix $\boldsymbol{D}_{ij} = \sum_k \boldsymbol{A}_{ik} \delta_{ij}$. The proximity between nodes $u$ and $v$ is then given by $\boldsymbol{\Pi}_{uv}^{\text{ppr}}$. Calculating this inverse is obviously infeasible. However, we can approximate $\boldsymbol{\Pi}^{\text{ppr}}$ with a sparse matrix $\tilde{\boldsymbol{\Pi}}^{\text{ppr}}$ in time $\mathcal{O}(\frac{1}{\varepsilon \alpha})$ per column, with error $\varepsilon \deg(u)$ (Andersen et al., 2006).

Next, we greedily construct the partition $P_{\text{I}}$ from $\tilde{\boldsymbol{\Pi}}^{\text{ppr}}$. To do so, we start by putting every node $u$ into a separate batch $\{u\}$. We then sort all elements in $\tilde{\boldsymbol{\Pi}}^{\text{ppr}}$ by magnitude, independent of their row or column. We scan over these values in descending order, considering the value's indices $(u, v)$ and merging the batches containing the two nodes. Finally, we randomly merge any small leftover batches. We stay within a memory constraint by only merging batches that stay below a maximum batch size. Note that the resulting partition is unbalanced. This method achieves well-overlapping batches and can efficiently add incrementally incoming primary nodes, e.g. in a streaming setting. Our experiments show that this method achieves a good compromise between well-overlapping batches and good gradients for training (see Sec. 5).

**Graph partitioning.** For our second method, we note that partitioning primary nodes into overlapping mini batches is closely connected to partitioning graphs. We can thus leverage the extensive amount of research on this topic by using the METIS graph partitioning algorithm (Karypis & Kumar, 1998) to find a partition of primary nodes $P_{\text{I}}$. Note that this approach completely skips step 1 in the above LBMB process. We found that graph partitioning yields roughly a two times higher overlap of auxiliary nodes than distance-based partitioning, thus leading to significantly more efficient batches. However, it also introduces a strong bias in batch selection that we found to be detrimental for training (see Sec. 5). Note that LBMB with graph partitioning is closely related to Cluster-GCN (Chiang et al., 2019). However, in contrast to Cluster-GCN our method ignores irrelevant parts of the graph and can obtain overlapping mini batches. This significantly accelerates training on small training sets and improves the accuracy of primary nodes close to the partition boundary.

## 3.2 Auxiliary node selection

**Generalizing influence scores.** The goal of selecting auxiliary nodes is to provide a neighborhood for each primary node that is as small as possible but still contains all relevant information. To quantify a node's importance, we can use the influence score of node $v$ on node $u$, $I(v, u) = \sum_i \sum_j \frac{\partial \boldsymbol{h}_{ui}}{\partial \boldsymbol{X}_{vj}}$. For GCN, this influence is proportional to a slightly modified random walk (Xu et al., 2018). We can thus select the most influential nodes for GCN by using this random walk landing probability. Unfortunately, different GNNs have different node influence scores, which are often even data-dependent.

To obtain a simple method that works with many different GNNs, we generalize the random walk landing probabilities obtained for GCN using *local clustering* methods. This is a well-established class of methods for finding a meaningful cluster for a selected node $u$ based on proximity in the graph. Examples include random walks with restart (or personalized PageRank (PPR), label propagation with return probability) (Andersen et al., 2006) and heat kernel (HK) diffusion (Kloster & Gleich, 2014). Local clustering only relies on the node's local neighborhood, making its runtime independent of the overall graph size and thus massively scalable. Since it is deterministic, we only need to perform its computation once during preprocessing. This allows us to use expensive methods such as $p$-norm flow diffusion (Fountoulakis et al., 2020), or methods that take node features into account. We leave these advanced methods for future work, and rely on basic PPR clustering in this work.

**Node-wise clustering.** Local clustering methods provide a separate cluster for each node, referred to as the "root node". For example, approximate PPR (Andersen et al., 2006) is guaranteed to provide all nodes with a PPR value $\boldsymbol{\Pi}_{uv}^{\text{ppr}} > \varepsilon \deg(v)$ w.r.t. the root node $u$. To create a batch of auxiliary nodes we can thus compute the local clusters of all primary nodes in a batch $\mathcal{S}_{\text{I}}$, and then merge them. If we partition the primary nodes according to their approximate PPR distances (distance-based partitioning) and select auxiliary nodes according to PPR clusters, we only need to calculate the PPR-scores once.

**Batch-wise clustering.** Considering each primary node separately does not take into account how one auxiliary node jointly affects multiple primary nodes. Fortunately, many

local clustering methods can be adapted to use a set of root nodes. For example, in PPR we can use a set of nodes in the teleport vector $\boldsymbol{t}$ instead of a single node, e.g. by leveraging the underlying recursive equation for a PPR vector $\pi_{\mathrm{ppr}}(\boldsymbol{t}) = (1 - \alpha)\boldsymbol{D}^{-1}\boldsymbol{A}\pi_{\mathrm{ppr}}(\boldsymbol{t}) + \alpha\boldsymbol{t}$. $\boldsymbol{t}$ is a one-hot vector in the node-wise setting, while for batched PPR it is $1/|\mathcal{S}_{\mathrm{I}}|$ for all nodes in $\mathcal{S}_{\mathrm{I}}$. This variant is also known as topic-sensitive PageRank. We found that batched PPR is significantly faster than node-wise clustering, especially in combination with partitioning the primary nodes via graph partitioning. However, it can lead to cases where one outlier node receives almost no neighbors, while others have excessively many. Whether node-wise or batch-wise clustering performs better thus often depends on the dataset and GNN.

### 3.3 Inference

In practice, a machine learning model is trained only once, while inference is run continuously once it is put into production. Even during training, inference is necessary for early stopping and performance monitoring. However, most previous GNN mini-batching methods only work well for training. These methods thus only have rather limited utility by themselves. Locality-based mini batching can be used for inference without any further changes, since it already focuses on the most important auxiliary nodes. It can therefore be used for all steps in the model's life cycle, or be combined with a separate method that focuses on training.

## 4 Gradient estimation with correlated batches

**Convergence.** Partitioning primary nodes based on proximity effectively correlates the gradients sampled in a batch. This might seem ill-motivated from the usual perspective of providing unbiased, low-variance gradient samples – sacrilegious even. However, we use every primary (training) node exactly once per epoch. The model thus sees all training labels equally often, ensuring an unbiased training process. Furthermore, there are multiple ways of mitigating the impact of correlated samples. Adaptive gradient descent methods such as Adam (Kingma & Ba, 2015) already cope rather well with imperfect gradient samples. We further improve this by adaptively reducing the learning rate when the validation loss plateaus, which ensures that the gradient step size decreases continuously. The resulting training scheme thus leads to convergence despite correlated samples, as shown in Sec. 5.

**Batch scheduling.** While we found Adam with learning rate scheduling to consistently ensure convergence, we still observed downward spikes in accuracy during training. To explain this issue, consider a sequence of multiple mini batches. In regular training every mini batch is similar and the order of these batches is irrelevant. In our case, however, some of the mini batches might be very similar. If the optimizer sees multiple similar batches in a row, it will do increasingly large steps in a suboptimal direction, which can lead to sporadic downward spikes in accuracy. To improve convergence we should therefore prevent such sequences of similar batches. To do so, we first quantify batch similarity using the normalized training class label distribution $p_i = c_i / \sum_j c_j$, where $c_i$ is the number of training nodes of class $i$. We then compare these distributions pairwise using the symmetrized KL-divergence, resulting in a pairwise batch distance $d_{ab}$ between batches $a$ and $b$. Based on this, we propose two ways of improving the batch schedule: 1. Find the fixed batch cycle that *maximizes* the batch distances between consecutive batches. This is a traveling salesman problem for finding the maximum distance loop that visits all batches. It is therefore only feasible for a small number of batches and otherwise must be approximated. 2. Sample the next batch weighted by the distance to the current batch. Both batch scheduling methods improve convergence and increase final accuracy, at almost no computational cost during training.

## 5 Experiments

**Experimental setup.** We primarily evaluate two variants of our method: LBMB with PPR distance-based batches and node-wise PPR clustering, and LBMB with graph partition-based batches and batch-wise PPR clustering. Both variants run in linear time $\mathcal{O}(N + E)$. We compare them to four state-of-the-art mini-batching methods: Neighbor sampling (Hamilton et al., 2017), Layer-Dependent Importance Sampling (LADIES) (Zou et al., 2019),

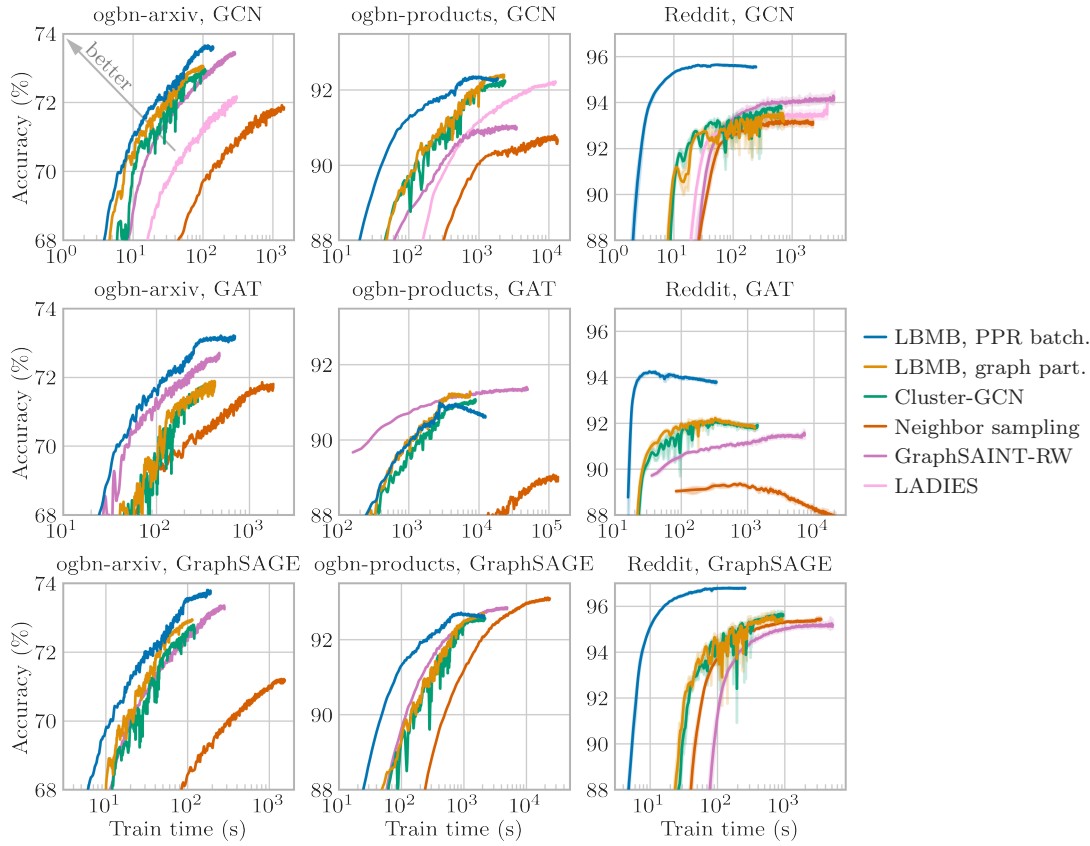

Figure 2: Convergence of validation accuracy in log. time. Average and 95 % confidence interval of 10 runs. LBMB converges the fastest in 8 of 9 cases.

GraphSAINT-RW (Zeng et al., 2020), and Cluster-GCN (Chiang et al., 2019). We use four large node classification datasets for evaluation: ogbn-arxiv (Hu et al., 2020; Wang et al., 2020, ODC-BY), ogbn-products (Wang et al., 2020, Amazon license), Reddit (Hamilton et al., 2017), and ogbn-papers100M (Hu et al., 2020; Wang et al., 2020, ODC-BY). While these datasets use the transductive setting, LBMB makes no assumptions about this and can equally be applied to the inductive setting. We skip the common, small datasets (Cora, Citeseer, PubMed) since they are ill-suited for evaluating scalability methods. We do not strive to set a new accuracy record but instead aim for a consistent, fair comparison based on three standard GNNs: graph convolutional networks (GCN) (Kipf & Welling, 2017), graph attention networks (GAT) (Veličković et al., 2018), and GraphSAGE (Hamilton et al., 2017). We use the same training pipeline for all methods, giving them access to the same optimizations. We run each experiment 10 times and report the mean and standard deviation in all tables and the bootstrapped mean and 95 % confidence intervals in all figures. See App. B for further details.

**Training.** To evaluate training performance we compare how fast the validation accuracy convergences for each method. We use LBMB inference since full inference is too slow to execute every epoch, and other approximate methods are either prohibitively slow or less accurate than LBMB. This already demonstrates a major advantage of LBMB: It is fast and accurate enough to run inference regularly *during* training. Fig. 2 shows how the accuracy increases depending on training time. LBMB performs significantly better than previous methods in 8 of 9 cases, converging up to 10x faster. This is *despite* the fact that we always prefetch the next batch in parallel. Note that GAT is much more computationally heavy than GCN and GraphSAGE, limiting the positive impact of a fast batching method. Computation-constrained models are less relevant in practice since data access is typically the bottleneck for real-world, disk-based

Table 1: Final accuracy and runtime averaged over 10 runs, with standard deviation. "Same method" refers to using the training method for inference, while "full graph" uses the whole graph for inference. LBMB achieves similar accuracy as previous methods when used for training, while using significantly less time per epoch and without requiring expensive full graph inference. LBMB is up to 500x faster than using the full graph for inference, at comparable accuracy. Other inference methods are substantially slower or less accurate. Note that LADIES is incompatible with the self loops in GAT and GraphSAGE.

| Setting | Training method | Time (s) | | | Test accuracy (%) | |
|---|---|---|---|---|---|---|
| | | Preprocess | Per epoch | Inference | Same method | Full graph |
| ogbn-arxiv, GCN | Full graph | - | - | 2.8 | - | - |
| | Neighbor sampling | **0.3** | 4.7 | 2.5 | 70.7±0.2 | 71.3±0.4 |
| | LADIES | **0.3** | 0.62 | 0.69 | 71.7±0.2 | 71.4±0.3 |
| | GraphSAINT-RW | 0.4 | 0.42 | 0.34 | 68.1±0.2 | *72.3±0.2* |
| | Cluster-GCN | 8.7 | **0.14** | *0.14* | 72.0±0.1 | *72.2±0.1* |
| | LBMB, graph part. | 14.1 | **0.14** | **0.13** | *72.2±0.2* | *72.2±0.2* |
| | LBMB, PPR batching | 17.5 | 0.27 | 0.16 | **72.7±0.1** | **72.7±0.1** |
| ogbn-arxiv, GAT | Full graph | - | - | 9.4 | - | - |
| | Neighbor sampling | **0.3** | 4.1 | 1.97 | *70.9±0.1* | *72.1±0.1* |
| | GraphSAINT-RW | *0.4* | 1.2 | 0.38 | 68.7±0.2 | **72.6±0.1** |
| | Cluster-GCN | 7.6 | *0.69* | **0.28** | 69.7±0.3 | 71.6±0.2 |
| | LBMB, graph part. | 7.7 | **0.68** | *0.31* | *71.0±0.3* | 71.8±0.3 |
| | LBMB, PPR batching | 17.6 | 1.53 | 0.98 | **72.2±0.1** | *72.3±0.2* |
| ogbn-arxiv, GraphSAGE | Full graph | - | - | 2.37 | - | - |
| | Neighbor sampling | **0.3** | 3.44 | 1.67 | 71.1±0.1 | 72.0±0.1 |
| | GraphSAINT-RW | **0.3** | 0.41 | 0.35 | 69.0±0.1 | **72.2±0.1** |
| | Cluster-GCN | 8.8 | **0.15** | *0.14* | 71.7±0.1 | 72.1±0.1 |
| | LBMB, graph part. | 7.2 | **0.15** | **0.13** | *72.0±0.2* | 72.1±0.1 |
| | LBMB, PPR batching | 17.5 | 0.31 | 0.15 | **72.4±0.2** | **72.4±0.2** |
| ogbn-products, GCN | Full graph | - | - | 130 | - | - |
| | Neighbor sampling | **32** | 42 | 433 | **78.2±0.2** | *78.0±0.2* |
| | LADIES | *33* | 25 | 22.5 | 75.9±0.3 | **79.0±0.4** |
| | GraphSAINT-RW | 35 | 11 | 20.6 | 52.6±0.8 | 77.0±0.3 |
| | Cluster-GCN | 302 | *3.7* | *3.4* | 76.2±0.3 | 76.5±0.2 |
| | LBMB, graph part. | 306 | **3.5** | **3.1** | 76.8±0.2 | 77.2±0.3 |
| | LBMB, PPR batching | 382 | 5.5 | 14.1 | *77.5±0.2* | *77.6±0.2* |
| ogbn-products, GAT | Full graph | - | - | 1700 | - | - |
| | Neighbor sampling | **33** | 450 | 3450 | **79.1±0.3** | 77.2±0.5 |
| | GraphSAINT-RW | *35* | 140 | 102 | 69.5±0.1 | **80.8±0.2** |
| | Cluster-GCN | 626 | **24** | *10.6* | 76.6±0.4 | 78.1±0.5 |
| | LBMB, graph part. | 767 | *25* | **10.0** | 77.0±0.4 | 78.9±0.6 |
| | LBMB, PPR batching | 378 | 41 | 94 | **79.3±0.3** | *79.4±0.3* |
| ogbn-products, GraphSAGE | Full graph | - | - | 88.0 | - | - |
| | Neighbor sampling | **31.4** | 52.0 | 530 | **81.0±0.2** | **81.4±0.2** |
| | GraphSAINT-RW | *35.8* | 10.6 | 20.0 | 69.4±0.2 | **81.3±0.2** |
| | Cluster-GCN | 313 | *3.1* | *3.4* | 79.5±0.4 | 79.7±0.4 |
| | LBMB, graph part. | 319 | **2.9** | **3.1** | 79.2±0.3 | 79.5±0.3 |
| | LBMB, PPR batching | 374 | 5.0 | 13.1 | *80.5±0.3* | 80.7±0.3 |
| Reddit, GCN | Full graph | - | - | 14.8 | - | - |
| | Neighbor sampling | **14.4** | 7.3 | 3.3 | 93.5±0.1 | 94.8±0.1 |
| | LADIES | *15.4* | 11.4 | 11.4 | *95.5±0.0* | *95.3±0.0* |
| | GraphSAINT-RW | 17.1 | 14.6 | 2.9 | 93.2±0.1 | **95.6±0.0** |
| | Cluster-GCN | 175 | 1.8 | 1.6 | 93.7±0.2 | 94.8±0.1 |
| | LBMB, graph part. | 175 | *1.6* | *1.4* | 93.5±0.4 | 94.7±0.1 |
| | LBMB, PPR batching | 64.8 | **0.72** | **0.57** | **95.7±0.0** | *95.2±0.0* |
| Reddit, GAT | Full graph | - | - | 76.9 | - | - |
| | Neighbor sampling | **14.8** | 70 | 32.5 | **94.3±0.1** | *95.1±0.1* |
| | GraphSAINT-RW | *17.9* | 21 | 3.2 | 79.4±0.2 | **95.4±0.1** |
| | Cluster-GCN | 366 | 4.7 | 1.4 | 91.4±0.1 | 93.5±0.7 |
| | LBMB, graph part. | 396 | *4.3* | *1.2* | 91.6±0.1 | 92.8±1.1 |
| | LBMB, PPR batching | 65.3 | **1.1** | **0.25** | **94.2±0.2** | 94.2±0.1 |
| Reddit, GraphSAGE | Full graph | - | - | 17.3 | - | - |
| | Neighbor sampling | **16.1** | 7.5 | 3.5 | *96.2±0.0* | **96.8±0.0** |
| | GraphSAINT-RW | *18.2* | 14.6 | 3.6 | 95.9±0.0 | **96.8±0.0** |
| | Cluster-GCN | 173 | 1.7 | 1.8 | 95.5±0.2 | 96.0±0.1 |
| | LBMB, graph part. | 175 | *1.6* | 1.7 | 95.6±0.2 | 96.1±0.1 |
| | LBMB, PPR batching | 66.0 | **0.77** | **0.65** | **96.8±0.1** | 96.4±0.0 |
| papers 100M, GCN | Full graph | - | - | 5700 | - | - |
| | Neighbor sampling | 739 | *900* | 159 | 64.3±0.2 | 61.8±0.2 |
| | LADIES | 735 | 2830 | 672 | *65.4±0.2* | *62.4±0.4* |
| | LBMB, PPR batching | 1160 | **104** | **11.7** | **66.0±0.1** | **66.3±0.0** |

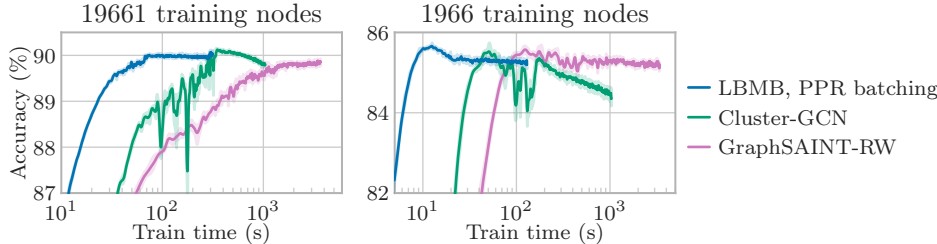

Figure 3: Training convergence in log. time for GCN on ogbn-products with smaller training sets. The gap in convergence speed between LBMB and the baselines grows larger for small training sets, since LBMB scales with training set size and not with overall graph size.

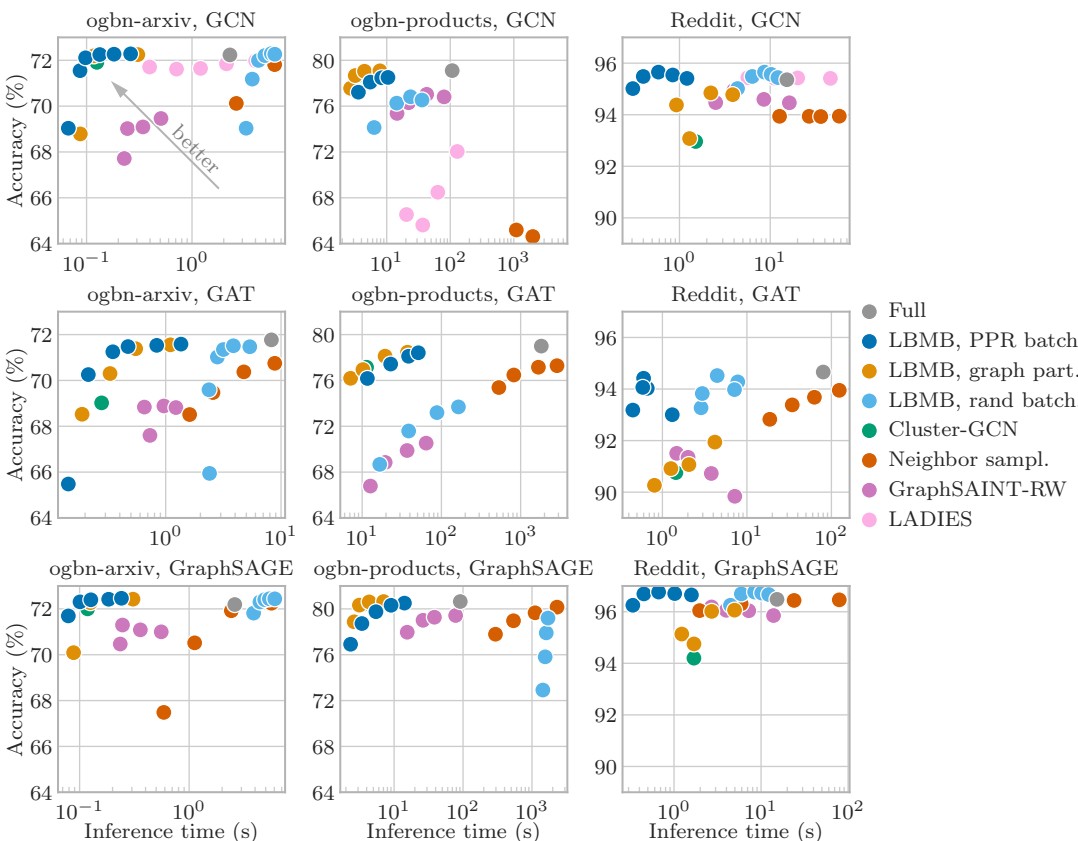

Figure 4: Test accuracy and log. inference time for a fixed GNN. LBMB consistently provides the best accuracy versus time trade-off (top-left corner).

datasets (Bojchevski et al., 2020). Table 1 furthermore shows that LBMB's time per epoch is significantly faster than all sampling-based methods. Cluster-GCN has a comparable runtime, which is expected due to its similarity with LBMB. However, it converges more slowly than LBMB with PPR batching and reaches a substantially lower final accuracy. Neighbor sampling achieves good final accuracy, but is extremely slow. GraphSAINT-RW only achieves good final accuracy with prohibitively expensive full graph inference. LBMB with PPR batching achieves the best final accuracy with a scalable inference method in 8 out of 10 cases. LBMB requires more preprocessing than previous methods. However, since LBMB is rather insensitive to hyperparameter choices (see Table 7, Fig. 10), preprocessing rarely needs to be re-run. Instead, its result can be saved to disk and re-used for training different models. Just considering our 10 training seeds, preprocessing of LBMB (PPR batching) only took 1.3 %

of the training time for GCN and $0.25\,\%$ for GAT on ogbn-arxiv. In some cases, LBMB uses more main memory than previous methods due to overlapping batches. However, it can also reduce memory requirements because it ignores irrelevant parts of the graph (see Table 6). LBMB with GCN outperforms SIGN-XL $((66.1\pm0.2)\,\%)$ (Frasca et al., 2020) on ogbn-papers100M, without any hyperparameter tuning and despite using 30x fewer parameters. LBMB furthermore has a substantially faster time per epoch and lower memory consumption than previous methods on this massive dataset, demonstrating LBMB's favorable scaling with dataset size. Notably, we were unable to evaluate GraphSAINT-RW and Cluster-GCN on this dataset, since they use more than $256\,\mathrm{GB}$ of main memory. Despite its fixed, correlated batches LBMB even performs similarly well as previous methods in terms of convergence per epoch (see Fig. 5). This demonstrates the importance of focusing on computational efficiency.

**Training set size.** The ogbn-arxiv and ogbn-products datasets both contain a large number of training nodes (91k and 197k, respectively). However, labeling training samples is often an expensive endeavor, and models are commonly trained with only a few hundred or thousand training samples. GraphSAINT-RW and Cluster-GCN are global training methods, i.e. they always use the full graph for training. They are thus ill-suited for the common setting of a large overall graph containing a small number of training nodes (resulting in a small label rate). In contrast, the training time of LBMB purely scales with the number of training nodes. To show this, we reduce the label rate by sub-sampling the training nodes of ogbn-products and compare the convergence in Fig. 3. As expected, the gap in convergence speed between LBMB and both Cluster-GCN and GraphSAINT-RW grows even larger for smaller training sets. Final test accuracies of all batching methods remain comparable in this setting.

**Inference.** Fig. 4 compares the inference accuracy and time of different batching methods, using the same pretrained model and varying computational budgets (number of auxiliary nodes/sampled nodes) at a fixed GPU memory budget. LBMB consistently provides the best trade-off between accuracy and time. PPR-based batching mostly performs better, except on ogbn-products, where it performs slightly worse than graph partitioning. LBMB provides a significant speedup over full graph inference, being 10 to 300 times faster at comparable accuracy. All previous methods are either significantly slower or less accurate.

**Ablation studies.** We ablate our primary node partitioning schemes by instead batching together random sets of nodes. We use fixed batches since we found that resampling incurs significant overhead without benefits – which is consistent with our considerations on gradient samples and contiguous memory accesses. Fig. 7 shows that this method ("Fixed random") converges more slowly and does not reach the same level of accuracy as our partition schemes. Fig. 4 shows that it ("LBMB, random batch.") is also substantially slower and often less accurate for inference. This is due to the synergy effects of primary node partitioning: If primary nodes have similar auxiliary nodes, they benefit from each other's neighborhood. We test auxiliary node selection by comparing LBMB to Cluster-GCN, since this just uses the graph partition as a batch instead of smartly selecting auxiliary nodes. We use the graph partition size as the number of auxiliary nodes for LBMB with graph partitioning to allow for a direct comparison. As discussed above, Cluster-GCN consistently performs worse, especially in terms of final accuracy, for inference, and for small label rates. Finally, Fig. 9 compares the proposed batch scheduling methods. Optimal and weighted sampling-based scheduling improve convergence and prevent or reduce downward spikes in accuracy. We explore further LBMB variants and hyperparameter choices in App. C.

## 6 Conclusion

We propose locality-based mini batching (LBMB), a method for extracting batches for GNNs. Unlike previous methods, we treat primary (output) and auxiliary nodes separately and focus on the computational aspects of training. LBMB can be used both for training and inference, and outperforms previous methods on both tasks. It improves time per epoch by up to 20x and inference time by up to 100x compared to previous methods that reach a similar accuracy. These improvements grow even larger in the common setting of sparse labels and when the pipeline is constrained by data access speed.

**Reproducibility.** We publish a reference implementation along with this paper for reproducibility. Additionally, App. B contains all hyperparameters, experimental settings, software versions, and the hardware configuration necessary for reproducing our results.

**Ethical considerations.** Scalable graph-based methods can enable the fast analysis of huge datasets with billions of nodes. While this has many positive use cases, it also has obvious negative repercussions. It can enable mass surveillance and the real-time analysis of whole populations and their social networks. This can potentially be used to detect emerging resistance networks in totalitarian regimes, thus suppressing chances for positive change. Voting behavior is another typical application of network analysis: Voters of the same party are likely to be connected to one another. Scalable GNNs can thus influence voting outcomes if they are leveraged for targeted advertising.

The ability of analyzing whole populations can also have negative personal effects in fully democratic countries. If companies determine credit ratings or college admission based on connected personal data, a person will be even more determined by their environment than they already are. Companies might even leverage the obscurity of complex GNNs to escape accountability: It might be easy to reveal the societal effects of your housing district, but unraveling the combined effects of your social networks and digitally connected behavior seems almost impossible. Scalable GNNs might thus make it even more difficult for individuals to escape the attractive forces of the status quo.

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

## A RELATED WORK

**Scalable GNNs.** Multiple works have proposed massively scalable GNNs that leverage the peculiarities of message passing to limit the model to a single message passing step, akin to label or feature propagation (Bojchevski et al., 2020; Frasca et al., 2020). In our work we are interested in general, model-agnostic scalability methods.

**Scalable graph learning.** Classical graph learning faced issues similar to GNNs when scaling to large graphs. Multiple frameworks for distributed graph computations were proposed to solve this without approximations or sampling (Gonzalez et al., 2012; Low et al., 2012; Malewicz et al., 2010; Kyrola et al., 2012). Other works scaled to large graphs via stochastic variational inference, e.g. by sampling nodes and node pairs (Gopalan et al., 2012). Interestingly, this approach is quite similar to sampling-based mini batching for GNNs.

**Mini batching for GNNs.** Previous mini-batching methods can be divided into three categories: Node-wise sampling, layer-wise sampling, and subgraph-based sampling (Liu et al., 2021). In node-wise sampling, we obtain a separate set of auxiliary nodes for every primary node, which are sampled independently for each message passing step. Each primary node is treated independently; if two primary nodes sample the same auxiliary node, we compute its embedding twice (Hamilton et al., 2017; Ying et al., 2018; Liu et al., 2020). Layer-wise sampling jointly considers all primary nodes of a batch to compute a stochastic set of activations in each layer. Computations on auxiliary nodes are thus shared (Chen et al., 2018; Huang et al., 2018; Zou et al., 2019). Subgraph-based sampling selects a meaningful subgraph and then runs the GNN on this subgraph as if it were the full graph. This method thus computes the outputs and intermediate embeddings of all nodes in that subgraph (Chiang et al., 2019; Zeng et al., 2020). Our method most closely resembles the subgraph-based sampling approach. We also use the full subgraph induced by the selected nodes and calculate the intermediate embeddings for all nodes in a batch. However, we only calculate the outputs of primary nodes, similar to node-wise sampling.

## B MODEL AND TRAINING HYPERPARAMETERS

**Hardware.** All experiments are run on an NVIDIA GeForce GTX 1080Ti. The experiments on ogbn-arxiv and ogbn-products use up to 64 GB of main memory. The experiments on ogbn-papers100M use up to 256 GB.

**Packages.** Our experiments are based on the following packages and versions:

- torch-geometric 1.7.0
  - torch-cluster 1.5.9
  - torch-scatter 2.0.6
  - torch-sparse 0.6.9
- python 3.7.10
- ogb 1.3.1
- torch 1.8.1
- cudatoolkit 10.2.89
- numba 0.53.1
- python-tsp 0.2.0

**Preprocessing.** Before training, we first make the graph undirected, and add self-loops. The adjacency matrix is symmetrically normalized. We cache the symmetric adjacency matrix for graph partitioning and mini-batching. Instead of re-calculating the adjacency matrix normalization factors for GCN for each mini batch, we re-use the global normalization factors. We found this to achieve similar accuracy at lower computational cost.

**Models.** We use three models for all the experiments: GCN (3 layers, hidden size 256 for the ogbn datasets and 2 layers, hidden size 512 for Reddit), GAT (3 layers, hidden size 128, 4 heads for the ogbn datasets and 2 layers, hidden size 64, 4 heads for Reddit), and GraphSAGE (3 layers, hidden size 256). All models use layer normalization, ReLU activation functions, and dropout. We performed a grid search on ogbn-arxiv, ogbn-products, and

Table 2: Number of batches for graph partition-based batching.

| Model | Dataset | Number of batches | | |
|-------|---------|-------|------------|------|
| | | Train | Validation | Test |
| GCN | ogbn-arxiv | 4 | 2 | 2 |
| GCN | ogbn-products | 16 | 8 | 8 |
| GCN | Reddit | 8 | 4 | 4 |
| GAT | ogbn-arxiv | 8 | 4 | 4 |
| GAT | ogbn-products | 1024 | 512 | 512 |
| GAT | Reddit | 400 | 200 | 200 |
| GCN | ogbn-papers100M | 256 | 32 | 48 |

Reddit to obtain the optimal model hyperparameters based on final validation accuracy. For ogbn-papers100M we use the same hyperparameters as for GCN on ogbn-arxiv, but with 32 auxiliary nodes per primary node.

**Training.** To minimize data loading and memory access overhead we always prefetch the next batch in parallel. We found that using more than one worker for data loading does not improve runtime, since loading is memory-bound, not compute-bound. We use the Adam optimizer for all the experiments, with a starting learning rate of $10^{-3}$. We use an $L_2$ regularization of $10^{-4}$ for GCN on ogbn-arxiv and ogbn-products, and no $L_2$ regularization in all other settings. We use a ReduceLROnPlateau scheduler for the optimizer, with the decay factor 0.33, patience 30, minimum learning rate $10^{-4}$, and cooldown of 10, based on validation loss. We train for 300 to 800 epochs and stop early with a patience of 100 epochs, based on validation loss. We determine the optimal batch order for LBMB via simulated annealing (Dréo et al., 2006).

**Graph partition-based batching.** We tune the number of batches and thus the size of batches using a grid search (see Table 2). Generally, final accuracy increases with larger batch sizes, but this can lead to excessive memory usage and slower convergence speed. Note that the inference batch size is double the sizes of training batches since in this case we do not need to store any gradients.

**PPR-based batching.** For PPR-based batching we first calculate the PPR scores for each primary node, and then pick the top-k nodes for each primary node as its auxiliary nodes. Generally we use the same batch size, i.e. number of nodes in a batch, as in graph partition-based batching, to keep the GPU memory usage similar. However, if the graph is too dense, we might have to increase the batch size of PPR-based batching, because it tends to create sparser batches. Note that the resulting number of batches might differ from Table 2. We tune the number of auxiliary nodes per primary node using a logarithmic grid search using factors of 2. Based on this we use 16 neighbors for ogbn-arxiv, 64 for ogbn-products and 8 for Reddit. Note that the number of auxiliary nodes is the main degree of freedom in LBMB. It influences preprocessing time, runtime, memory usage, and accuracy. The number of primary nodes per batch is then determined by the available GPU memory.

**Random batching.** Random batching is similar to PPR-based batching except that the auxiliary nodes are batched randomly. We first calculate the PPR scores and pick the top-k neighbors as the auxiliary nodes for a primary node. We choose the same number of neighbors as with PPR-based batching. We investigate 2 variants of random batching: Resampling the batches in every epoch, and sampling them once during preprocessing and then fixing the batches. We only show the results for the second method, since we found it to be significantly faster, albeit requiring significantly more main memory.

**Hyperparameter tuning.** The priorities for tuning the hyperparameters are as follows: 1. To keep methods comparable in a realistic setup, we keep the GPU memory usage constant between methods. 2. When there are semantic hyperparameters that do not influence performance (such as the number of steps per epoch in GraphSAINT-RW, which only changes how an epoch is defined), we choose them to be comparable to other methods. 3. We choose all relevant hyperparameters based on validation accuracy. We use this process for both LBMB and the baselines.

Table 3: Hyperparameters for LADIES

| Model | Dataset | Nodes per layer | |
|---|---|---|---|
| | | Train | Validation |
| GCN | ogbn-arxiv | 42 336 | 84 672 |
| GCN | ogbn-products | 204 085 | 306 128 |
| GCN | Reddit | 90 000 | 150 000 |

Table 4: Hyperparameters for neighbor sampling

| Model | Dataset | Number of batches | | | Number of nodes |
|---|---|---|---|---|---|
| | | Train | Validation | Test | |
| GCN | ogbn-arxiv | 12 | 8 | 8 | 6, 5, 5 |
| GCN | ogbn-products | 20 | 4 | 200 | 5, 5, 5 |
| GCN | Reddit | 8 | 4 | 4 | 12, 12 |
| GAT | ogbn-arxiv | 8 | 4 | 4 | 8, 7, 5 |
| GAT | ogbn-products | 1000 | 150 | 8000 | 15, 10, 10 |
| GAT | Reddit | 400 | 400 | 400 | 20, 20 |

**Baseline hyperparameters.** For Cluster-GCN the number of batches are the same as for our graph partition-based batching variant. Table 3 shows the hyperparameters for LADIES, Table 4 for neighbor sampling, and Table 5 for GraphSAINT-RW. To ensure that every node is visited exactly once during GraphSAINT-RW inference we use the validation/test nodes only as root nodes of the random walks.

**Full graph inference.** We chunk the adjacency matrix and feature matrix for full-graph inference to allow using the GPU even for larger datasets. The only hyperparameter is the number of chunks. We limit the chunk size to ensure that full-graph inference does not exceed the amount of GPU memory used during training.

Table 5: Hyperparameters for GraphSAINT-RW

| Model | Dataset | Walk length | Sample coverage | Number of steps | Batch size | |
|---|---|---|---|---|---|---|
| | | | | | Train | Val/Test |
| GCN | ogbn-arxiv | 2 | 100 | 4 | 25 000 | 10 000 |
| GCN | ogbn-products | 2 | 100 | 16 | 80 000 | 5000 |
| GCN | Reddit | 2 | 100 | 8 | 23 000 | 6000 |
| GAT | ogbn-arxiv | 2 | 100 | 8 | 17 500 | 10 000 |
| GAT | ogbn-products | 2 | 100 | 1024 | 14 000 | 100 |
| GAT | Reddit | 2 | 100 | 400 | 1600 | 60 |

## C ADDITIONAL EXPERIMENTAL RESULTS

**Main memory usage.** Since LBMB saves the preprocessed dataset to memory, it can have a rather different main memory footprint than the normal dataset. To illustrate this, consider the Reddit dataset. LBMB uses 8 auxiliary nodes per primary node on this dataset, and the train and validation sets of Reddit contain 177k nodes (of a total of 233k nodes). If there was no overlap, this would translate to 1.4 million auxiliary nodes. However, our method aims at increasing precisely this overlap between auxiliary nodes in a batch (see Sec. 3). With graph partitioning, we end up with 466k auxiliary nodes, and with PPR batching we have 416k auxiliary nodes, across all batches. The number of auxiliary nodes determines the size of the batched feature matrix and thus the memory size. The number of edges per batch also impacts memory usage, but the feature matrix is usually the dominant factor. The edges do, however, have a major impact on GNN computation time. Overall, we should expect a memory usage of around $416/233 = 178\%$ with PPR batching, and $466/233 = 200\%$ with graph partitioning, compared to the normal dataset. All in all, LBMB's memory usage thus depends on three aspects: 1) How large is the training/validation set compared to the full graph? 2) How many auxiliary nodes per primary node are we using? 3) How well are the batches overlapping?

So why are we not seeing a memory overhead in the measured usage for Reddit in Table 6? Because LBMB requires no memory for sampling. Constructing and prefetching batches requires a surprisingly large amount of memory, especially if you aim to saturate main memory or PCIe bandwidth ( 30GB/s). The memory used by sampling is required in addition to the normal dataset, while LBMB can delete the dataset from memory after preprocessing.

**LBMB variants.** Fig. 7 shows that PPR-based batching converges faster than both graph partition-based and random batching. This suggests that the middle ground between unbiased gradient samples (random batching) and strongly overlapping batches (graph partitioning) is beneficial. Furthermore, graph partition-based batching converges faster than fixed random batching in time, but not per epoch. This demonstrates the impact of auxiliary node overlap on runtime. We found that LBMB is largely insensitive to different local clustering methods and hyperparameters for selecting auxiliary nodes (see Table 7). Fig. 10 shows that the batch size (number of primary nodes per batch) also only has a minor influence on accuracy, especially above 1000 primary nodes per batch. Consequently, in practice LBMB only has a single free hyperparameter to choose: The number of auxiliary nodes per primary node. This can be determined in a few training runs. The number of primary nodes per batch is then given by saturating the available GPU memory, and the local clustering method and hyperparameters are not important. Note that Fig. 10 shows that LBMB performs surprisingly well with small batches. LBMB can thus even be used in extremely constrained settings with small batches of 100 primary nodes per batch.

Table 6: Main memory usage (GiB). In some cases, LBMB uses more main memory than previous methods due to overlapping batches (e.g. on ogbn-products). However, it can also reduce memory requirements because it ignores irrelevant parts of the graph (e.g. on Reddit). Note that we chose hyperparameters in a way that keeps GPU memory usage roughly constant between methods (as opposed to main memory usage).

| | ogbn-arxiv | | | ogbn-products | | | Reddit | | |
|---|---|---|---|---|---|---|---|---|---|
| | GCN | GAT | GraphSAGE | GCN | GAT | GraphSAGE | GCN | GAT | GraphSAGE |
| Neighbor sampling | 3.0 | 3.6 | 3.1 | 8.7 | 7.9 | 8.5 | 7.4 | 7.5 | 7.1 |
| LADIES | 3.0 | - | - | 6.0 | - | - | 4.8 | - | - |
| GraphSAINT-RW | 3.5 | 3.6 | 3.5 | 9.6 | 9.6 | 9.6 | 8.4 | 8.5 | 8.4 |
| Cluster-GCN | 3.5 | 3.4 | 3.5 | 7.8 | 6.0 | 7.3 | 6.1 | 4.2 | 6.5 |
| LBMB, graph part. | 3.5 | 3.6 | 3.5 | 7.9 | 7.0 | 7.8 | 6.3 | 4.9 | 6.3 |
| LBMB, PPR batching | 3.8 | 3.8 | 4.2 | 13.0 | 12.3 | 13.2 | 4.5 | 5.3 | 5.1 |

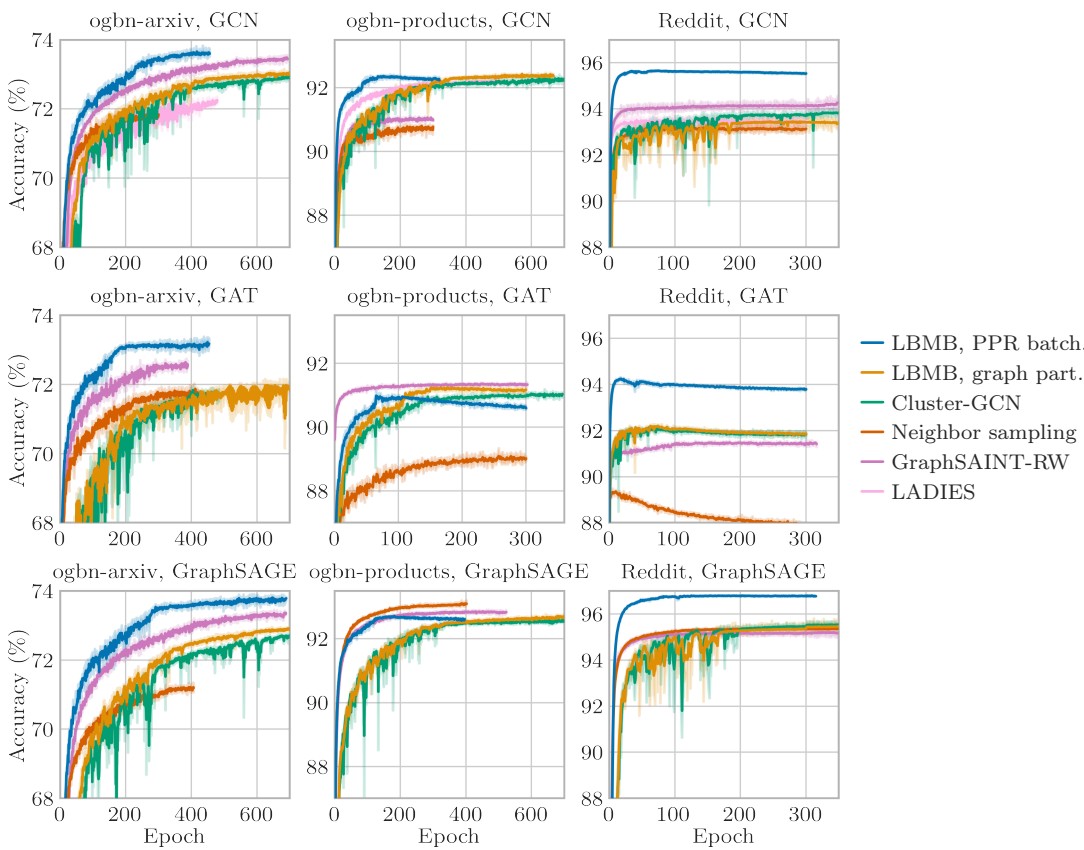

Figure 5: Convergence per epoch. LBMB converges similarly fast in most cases, despite not performing any sampling.

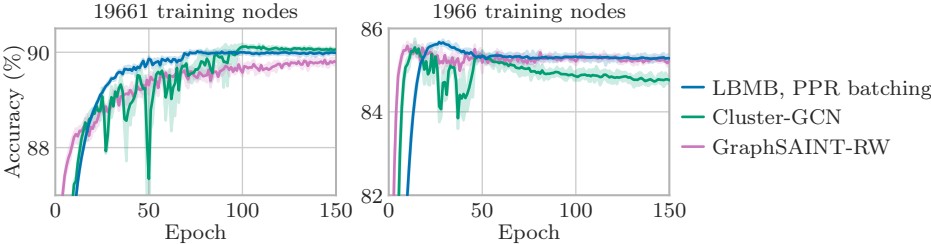

Figure 6: Training convergence per epoch for smaller training sets. All methods converge similarly fast per training step, demonstrating once again the importance of a fast time per training step.

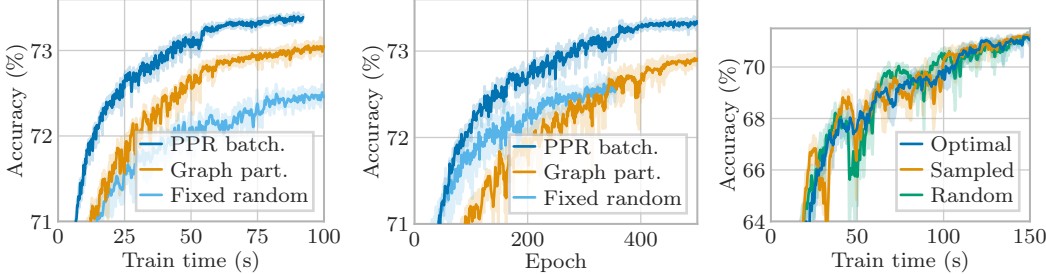

Figure 7: Convergence per time for training GCN on ogbn-arxiv. Both graph partitioning and PPR batching lead to faster convergence than fixed random batching.

Figure 8: Convergence per epoch for batching methods when training GCN on ogbn-arxiv. PPR-based partitioning also converges the fastest per training step.

Figure 9: Batch scheduling for GAT on ogbn-arxiv. Optimal batch order prevents downward spikes in accuracy and leads to higher final accuracy.

Table 7: Methods and hyperparameters for selecting auxiliary nodes for GCN on ogbn-products. LBMB is very robust to this choice. We did observe a slightly lower validation accuracy for a low alpha (0.05), so we recommend using 0.25.

| Method | $\alpha$, $t$ | Time (s) per epoch | Test accuracy (%) LBMB inference | Full graph |
|---|---|---|---|---|
| PPR | 0.05 | 3.5 | 76.8±0.3 | 77.1±0.3 |
| PPR | 0.15 | 3.6 | 76.6±0.4 | 76.9±0.4 |
| PPR | 0.25 | 3.5 | 76.8±0.2 | 77.2±0.3 |
| PPR | 0.35 | 3.5 | 76.9±0.5 | 77.2±0.5 |
| Heat kernel | 0.1 | 3.5 | 76.5±0.4 | 76.8±0.3 |
| Heat kernel | 1 | 3.5 | 76.6±0.5 | 76.9±0.5 |
| Heat kernel | 3 | 3.5 | 76.8±0.2 | 77.1±0.2 |
| Heat kernel | 5 | 3.5 | 76.7±0.5 | 77.0±0.5 |
| Heat kernel | 7 | 3.5 | 76.6±0.4 | 76.8±0.4 |

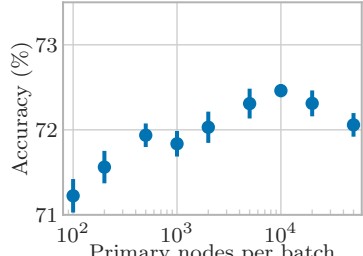

Figure 10: Final test accuracy (LBMB inference) when training LBMB with PPR batching with different numbers of primary nodes per batch (GCN on ogbn-arxiv). LBMB is rather insensitive to this choice.

# D  ADDITIONAL CONSIDERATIONS

**Experimental limitations.** We only tested our method on homophilic node classification datasets. While proximity is a central inductive bias in all GNNs, we did not explicitly test this on a more general variety of graphs. However, note that LBMB does *not* require homophily. The underlying assumption is merely that nearby nodes are the most important, not that they are similar. Finally, we expect our method to perform even better in the context of billion-node graphs, but our benchmark datasets still fit into main memory.

**Micro batching.** To further smoothen the gradients we could accumulate gradients across multiple batches before performing a gradient step, a method known as micro batching. This is especially well-suited for calculating the gradients of multiple batches in parallel, e.g. via distributed training. With some overhead, we can even include inter-batch edges in this case, as proposed by Cluster-GCN (Chiang et al., 2019). We found that neither regular micro batching nor parallel micro batching with inter-batch edges yields noticeable improvements for the datasets we considered.

**Exponential moving averages.** Another common method for smoothing noisy model weight updates are exponential moving averages (EMA) of the model parameters. This approach uses two sets of model parameters: The regular training parameters and the shadow parameters. The shadow parameters are updated in each step by a fraction of the current training parameters, thus giving an exponential moving average of the regular parameters. The shadow parameters are never used during training, only for validation and prediction. While this method often has a positive impact on accuracy, we did not use it for the sake of simplicity.

**Scalable batch scheduling.** The batch scheduling methods we introduced in Sec. 4 are quadratic in the number of batches, which might become problematic for massive datasets. While this was not a problem in our experiments, we can overcome this by scheduling based on batch clusters instead of individual batches.

**Long-range interactions.** Many models in deep learning have dozens or even hundreds of layers (He et al., 2016). However, modern GNNs typically only have a few layers. Intuitively, this might seem like a strong limitation, and is often attributed to oversmoothing (Li et al., 2018). However, GNNs that overcome oversmoothing still only rely on a close neighborhood (Xu et al., 2018; Klicpera et al., 2019a). This holds for models focusing on heterophilic graphs as well (Zhu et al., 2021b). Even structural (role-based) embeddings are based on a measure of proximity (Zhu et al., 2021a), which can also be exploited for mini batching. This difference to regular neural networks is most likely due to the message passing dynamics in GNNs. The dynamics are very different due to the permutation invariance required in aggregation. The aggregation mechanism in Eq. (2) means that nodes at a far distance are aggregated across multiple layers, and can thus no longer be represented individually (Alon & Yahav, 2021). Only very few models are capable of preserving messages over long distances (Beaini et al., 2021; Alon & Yahav, 2021), and none have demonstrated benefits for large networks. On the contrary, simple label propagation and diffusion methods can substantially improve the accuracy of GNNs on large graphs (Klicpera et al., 2019b; Huang et al., 2021). Overall, current evidence strongly suggests that in most datasets the close neighborhood is disproportionately more important for GNN predictions than distant neighbors. While the role of long-range interactions in GNNs still remains an open topic of research, focusing on local clusters thus does not appear to be a limitation for most GNNs. This is also evidenced by many previous scalability methods relying on locality as well (Zeng et al., 2020; Chiang et al., 2019).
For settings that still require long-range interactions, locality-based mini batching could be extended using special globally connected nodes, parallel micro batching with inter-batch edges (see Sec. 4), or steady-state learning (Dai et al., 2018) – possibly with compression to enable scaling to large graphs (Rae et al., 2020).

