# OpenReview forum: "Locality-Based Mini Batching for Graph Neural Networks"
_ICLR.cc/2022/Conference — ICLR 2022 Submitted_

### Official Review · Reviewer_AWEr · 2021-11-02

**Correctness:** 2
**Technical Novelty And Significance:** 2
**Empirical Novelty And Significance:** 3
**Recommendation:** 5
**Confidence:** 4

**Main Review:**

This work studies an important question that plagues the training and inference of graph neural networks in a large scale. Random memory access is not necessarily the root cause that prevents faster training, but indeed the extra cost to reorganize batch data into consecutive memory is an anchor of improvement. The authors precompute the batches and shift the burden from training to preprocessing; this is an interesting idea to explore.

Major issues:

Although generally it is argued that preprocessing is done only once and sounds less important, note that training (and inference) is also done only once for graph neural networks. Hence, a comprehensive study should also compare the total time in preprocessing plus training (possibly including also inference) across different methods. This is lacking in Table 1.

Another concern is the memory cost incurred to store the mini batches. A node may appear in multiple batches, causing multiple times storage increase compared to usual sampling methods. This problem is more pronounced when the training set is large (causing more overlaps among the batches). Certain analysis would be desirable.

A lot of technical details developed in the paper (such the the partitioning approaches and the approaches to selecting auxiliary nodes) follow straight intuition. The authors provide no theoretical analysis/support and they do not even prefer an approach, when multiple choices are equally plausible. This renders the paper rather empirical. Then, strong experiment results are needed to give a reader confidence to adopt the proposed techniques. However, the current empirical results are not sufficiently strong. For example, Table 1 does not appear to suggest an exclusively advantageous method. Moreover, the experiment section can be stronger if results of more networks (e.g., GraphSAGE and GIN) are demonstrated.

An important problem is that deterministic batches go entirely outside the realm of stochastic optimization, while also not being applicable to a usual deterministic optimization that requires the full gradient. Convergence and convergence rates are unclear and obviously the authors also face irregularities (such as "downward spikes" mentioned in Section 4). More in-depth study, from either the theoretical side or the empirical side, is needed to solidify the proposed work.

Minor issues/questions:

At the beginning of section 3, it is mentioned that the first step of LBMB obtains the $k$ most important auxiliary nodes for each primary node. However, section 3.1 discusses partitioning without the use of these auxiliary nodes. Could the authors clarify?

In table 1, preprocessing time for the papers100M data set is missing.

In table 1, is the inference time using "same method" or "full graph"?

Table 1 can benefit from also reporting the number of training epochs to reach the test accuracy listed thereof.

---

After rebuttal

The authors made good arguments regarding my initial concerns on expensive preprocessing time and costly memory consumption. I therefore adjust the score. There remain concerns regarding too many options (none preferred) in the proposed method and the unclear impact on training convergence. The idea by itself is illuminating and with polish and prudence in use, it may lead to a leap in the training of GNNs in a large scale.

**Summary Of The Paper:**

This paper studies the training of graph neural networks and proposes a mini batching scheme that circumvents the expensive cost of traditional sampling approaches due to random memory access. The authors construct deterministic mini batches so that they can be laid out in consecutive memory, which speeds up training. The authors empirically show that such a mini batching scheme significantly improves training and inference time, at the cost of precomputation.

**Summary Of The Review:**

This paper studies an important question. The proposed method, however, has multiple issues including timing, storage, convergence, and insufficient empirical evidence.

---

> ### Author Response · Authors · 2021-11-21
> **Similar memory usage, strong results for GraphSAGE, and why preprocessing time is secondary (part 2)**
>
> - **Empirical approach:** We agree that our contribution is largely empirical in nature. However, it is based on general considerations of computer architectures and theoretical insights on the properties of GNNs (e.g. influence scores). Furthermore, we embrace the empirical nature of our paper by presenting experiments that go far beyond the scope presented in previous works: 10 experimental setups, 5 methods, 10 repetitions per method. We only use large datasets and use a single, consistent pipeline that equally optimizes our method and all baselines. We present full convergence curves, inference measurements, timings of all steps, memory usage, and ablation studies. We have now also **added experiments for GraphSAGE**, as suggested. LBMB works even better with this model than with GCN or GAT.
> - **No single best method on all measures and settings:** We hope that the reviewer appreciates the complexity and diversity of models and graph datasets, which naturally leads to the fact that no single method can perform best in all settings on all measures -- if evaluated fairly. The same finding is present in many previous works, such as [2]. Still, LBMB converges faster in 7/9 settings (Figure 2), provides the best runtime vs. accuracy trade-off for inference in _all_ settings (Figure 4), and the best final test accuracy (training+inference) in 8/10 settings (Table 1). It achieves an up to 20x improvement in training convergence and 100x improvement in inference speed. We find these results to be quite strong.
> - **Deterministic batches:** We agree that our deterministic batching methods go outside the realm of what was explored previously, which shows our work's novelty. We show the full convergence plots for 23 experimental settings (models, datasets, hyperparameters) to clearly present the convergence and convergence rates. The mentioned "downward spikes" are also depicted in the paper (Figure 9) and solved by our batching schedules.
>
> Minor questions:
> - The explanation in Section 3 is a general, theoretical consideration on how to optimally solve the task, which also sets the framework for our method. Sections 3.1 and 3.2 then go from theory to practice and present methods that solve this in a scalable manner. The graph partitioning variant of LBMB circumvents the necessity of finding the k most important nodes for each primary node, by partitioning the graph directly.
> - We used a somewhat different method for preprocessing papers100M than for the other settings, as explained in the appendix. The time is therefore not comparable, and we decided to remove it.
> - The inference time is for "same method". Otherwise all times would be the same. The full graph inference time is listed in the "full graph" rows.
> - The number of training epochs usually depends on an arbitrary early stopping criterion, which would arbitrarily benefit one method over another (usually LBMB with PPR batching). If you are interested in this, please refer to Figure 5, where you can see the validation maxima of all methods.
>
> [1] https://d1.awsstatic.com/events/reinvent/2019/REPEAT_1_Deliver_high_performance_ML_inference_with_AWS_Inferentia_CMP324-R1.pdf
>
> [2] Oleksandr Shchur, Maximilian Mumme, Aleksandar Bojchevski, Stephan Günnemann. Pitfalls of Graph Neural Network Evaluation. NeurIPS-W 2018

---

> > ### Comment · Reviewer_AWEr · 2021-11-22
> > **Follow up the rebuttal**
> >
> > The authors' point on preprocessing vs. training makes sense.
> >
> > The inference time is quite comparable with the per-epoch training time (for graph neural networks). Hence, I am not sure if the argument with citation [1] is helpful here.
> >
> > The memory storage for the precomputed batches is indeed case dependent (on, e.g., the training set size and batch overlap), but it cannot be avoided, unless the batches are streamed in from disk on demand, in which case IO time needs to be factored in. Hence, a conceptual discussion of how this uniquely incurred memory consumption compares with that of other methods would be more helpful than listing a table obtained through experimental measurement on specific data sets.
> >
> > The additional results for GraphSAGE are appreciated.
> >
> > The problem of deterministic batches really needs more in-depth study; otherwise, using the proposed method feels like trying luck.
> >
> > Based on the authors' experience with preprocessing papers100M, there may be a potential scalability challenge, because the proposed method needs graph partitioning that cannot handle graphs of such a scale and beyond.
> >
> > It is still important to list the preprocessing time for papers100M (and explain that the number is incomparable).

---

> > > ### Author Response · Authors · 2021-11-23
> > > **Memory discussion, luck & determinism, and the preprocessing time for papers100M (part 2)**
> > >
> > > **Preprocessing on papers100M:**
> > > Thanks to our further optimized preprocessing pipeline we were now able to run papers100M with the regular PPR batching method (which does not use graph partitioning), without any special treatment. We have added preprocessing times to the table accordingly. We were not able to update all of our papers100M results (other times and accuracy) in the few hours between your response and the deadline, but will do so in the camera-ready version. **Our preprocessing time (1160s) is surprisingly similar to the simple preprocessing for neighbor sampling (739s)**, which only needs to add self loops and symmetrize and normalize the adjacency matrix. This shows the scalability of our method: PPR calculation is purely local and does not depend on the overall graph size. By the way, note that while METIS does not scale to graphs as large as papers100M, there are multiple other graph partitioning methods that do, see e.g. [2, 3, 4].
> > >
> > > [1] Bojchevski, Klicpera, Perozzi, Kapoor, Blais, Rozemberczki, Lukasik, Günnemann. Scaling Graph Neural Networks with Approximate PageRank. KDD 2020
> > >
> > > [2] Stanton, Kliot. Streaming Graph Partitioning for Large Distributed Graphs. KDD 2012
> > >
> > > [3] Tsourakakis, Gkantsidis, Radunovic, Vojnovic. FENNEL - Streaming Graph Partitioning for Massive Scale Graphs. WSDM 2014
> > >
> > > [4] Wang, Xiao, Shao, Wang. How to Partition a Billion-Node Graph. ICDE 2014

---

> > > > ### Comment · Reviewer_AWEr · 2021-11-23
> > > > **follow up**
> > > >
> > > > Some discussions in this two-part reply are valuable, such as those regarding memory consumption; they will make the paper more solid.
> > > >
> > > > While some other discussions are debatable, and it probably is not the point to have an endless debate here, I do want to suggest one problem for the authors to consider, which hopefully would increase the value of the work:
> > > >
> > > > If we formalize the deterministic batch idea into a mathematical problem, it is to say that we want to minimize a loss $f(w)$ where $f=\sum_i f_i(w)$, each $f_i$ corresponding to one deterministic batch. To make a simplified example, let us say $f(w)=g(w)+h(w)$, where $g(w)=w^2$ and $h(w)=w$. The minimizer is $w^*=-0.5$. However, if we do gradient steps alternating between $g$ and $h$ (that is, $w\gets w-\alpha p(w)$, where $\alpha$ is a fixed learning rate and $p$ is either $\nabla g$ or $\nabla h$), then the iterate $w$ eventually oscillates between $-0.5$ and $-0.5+\alpha$.
> > > >
> > > > There are two ways to resolve this problem. We could either keep alternating between $g$ and $h$, but add a consensus constraint so that the optimization converges to the right place; or do not immediately update the model after each batch. In the latter case, we accumulate the gradients for all batches, equivalently performing deterministic gradient descent. I suggest the authors try both strategies and see how the convergence compares with stochastic gradient descent.
> > > >
> > > > Of course, if we do the latter approach, we don't even need batching. Computing the full loss layer by layer probably would be faster.

---

> > > > > ### Author Response · Authors · 2021-11-25
> > > > > **Consensus constraints, adaptive optimization, and gradient accumulation**
> > > > >
> > > > > Thank you for engaging so consistently and constructively in this discussion!
> > > > >
> > > > > **Prior results on deterministic batching:**
> > > > > As a side note, we would first like to highlight that prior works have also found benefits in deterministically selecting mini batches [1, 2]. While these were focussed on constructing similar or well-representative mini batches, this still goes to show that deterministic batching often offers advantages.
> > > > >
> > > > > **Consensus constraints and adaptive optimization:**
> > > > > Using a consensus constraint between batches is a very interesting and insightful idea. To incorporate this, we state our problem as a distributed optimization setup with one processor per mini batch. We then add the consensus constraint that parameters between mini batches should be the same. This allows us to solve it using a distributed primal-dual saddle-point algorithm. Due to the directedness of time we use a directed communication graph between processors, leading to the (provably converging) dynamics [3]
> > > > > $$
> > > > > \dot{x} + \alpha \mathbf{L} x + \mathbf{L} z = - \nabla \tilde{f}(x),
> > > > > $$
> > > > > $$
> > > > > \dot{z} = \mathbf{L} x,
> > > > > $$
> > > > > with the weights $x$, the dual variable $z$, the communication graph's Laplacian $\mathbf{L} = \mathbf{D}_{\text{out}} - \mathbf{A}$, and some $\alpha \ge 2\sqrt{2}$. We connect every processor $t$ with its preceding one, leading to the dynamics
> > > > > $$
> > > > > \dot{x}^{(t)} + \alpha (x^{(t)} - x^{(t-1)}) + (z^{(t)} - z^{(t-1)}) = - \nabla \tilde{f}^{(t)}(x^{(t)}),
> > > > > $$
> > > > > $$
> > > > > \dot{z} = x^{(t)} - x^{(t-1)}.
> > > > > $$
> > > > > In discretized time we have $x^{(t)} = x^{(t-1)} + \lambda \dot{x}^{(t-1)}$, with the learning rate $\lambda$, and the same for $z^{(t)}$. This allows us to simplify the dynamics to obtain the update equation
> > > > > $$
> > > > > \dot{x}^{(t)} = - \nabla \tilde{f}^{(t)}(x^{(t)}) - \alpha \lambda \dot{x}^{(t-1)} - \lambda^2 \dot{x}^{(t-2)}.
> > > > > $$
> > > > > Now, recall that for SGD with momentum we have $\dot{x}^{(t)} = - \nabla \tilde{f}^{(t)}(x^{(t)}) - \beta \dot{x}^{(t-1)}$, which is strikingly similar to the above equation. Indeed, the above equation fits perfectly within the framework of adaptive optimization methods [4]. And if we consider a case that would normally cause oscillations (such as your example), we see that momentum and adaptive methods indeed suppress these oscillations, just as you suggested! Adagrad and Adam were even motivated by sparse gradients, such as the ones caused by our mini-batching scheme [5, 6]. In Section 4 we already mention the importance of using Adam for convergence. We will incorporate these considerations in the paper to further improve our explanation.
> > > > >
> > > > > **Gradient accumulation:**
> > > > > We have already performed experiments with gradient accumulation (also known as micro batching), but found that the improvements were only minor. So we opted for simplicity over performance (see Appendix D). You can find the associated figures [in this figshare](https://figshare.com/s/9a977b8334cd03be0d90). Note that we split the training nodes into 4 batches for GCN on ogbn-arxiv (see Table 2; note that other settings use up to 1024 batches). So using 4 micro batches amounts to using full-batch gradient descent, as suggested. Micro batching does indeed smoothen out the convergence curve. Overall, there is a minor improvement in convergence and final accuracy, with 2 micro batches performing best. We will add this plot to the paper.
> > > > >
> > > > > **Full-batch training:**
> > > > > Gradient checkpointing and matrix chunking can indeed sometimes allow us to calculate the full gradients layer-by-layer. Chunking is actually the method we use for full graph inference. In Table 1 you can see that this is orders of magnitude slower than mini batching methods. And gradient checkpointing adds even more overhead to this.
> > > > >
> > > > > [1] Wang, Bai, Lavania, Bilmes. Fixing Mini-batch Sequences with Hierarchical Robust Partitioning. AISTATS 2019
> > > > >
> > > > > [2] Banerjee, Chakraborty. Deterministic Mini-batch Sequencing for Training Deep Neural Networks. AAAI 2021
> > > > >
> > > > > [3] Gharesifard, Cortés. Distributed Continuous-Time Convex Optimization on Weight-Balanced Digraphs. IEEE Transactions on Automatic Control, Vol. 59, No. 3, March 2014.
> > > > >
> > > > > [4] Reddi, Kale, Kumar. On the Convergence of Adam and Beyond. ICLR 2018
> > > > >
> > > > > [5] Duchi, Hazan, Singer. Adaptive Subgradient Methods for Online Learning and Stochastic Optimization. Journal of Machine Learning Research 12 (2011), 2121-2159
> > > > >
> > > > > [6] Kingma, Ba. Adam: A Method for Stochastic Optimization. ICLR 2015

---

> > > ### Author Response · Authors · 2021-11-23
> > > **Memory discussion, luck & determinism, and the preprocessing time for papers100M (part 1)**
> > >
> > > Thank you for following up so promptly! We hope that these explanations answer your further questions and comments:
> > >
> > > **Inference vs. training:**
> > > Your statement is true for benchmarks used in the scientific community, since the train, validation, and test sets are similar in size (e.g. ogbn-arxiv 90941 train, 29799 val, 48603 test nodes). However, in practice the "test" set is much larger than the training set: Once a GNN is trained and put into production, it needs to run inference continually for days and weeks. It continually serves queries for millions or billions of users, with predictions being frequently updated based on the latest data. There are good reasons why Amazon, Google, and others have developed specialized hardware for inference, and GNNs are no exception.
> > >
> > > **Conceptual discussion on memory consumption:**
> > > We have added the following discussion to the paper:
> > >
> > > > **Main memory usage.** Since LBMB saves the preprocessed dataset to memory, it can have a rather different main memory footprint than the normal dataset. To illustrate this, consider the Reddit dataset. LBMB uses 8 auxiliary nodes per primary node on this dataset, and the train and validation sets of Reddit contain 177k nodes (of a total of 233k nodes). If there was no overlap, this would translate to 1.4 million auxiliary nodes. However, our method aims at increasing precisely this overlap between auxiliary nodes in a batch (see Section 3). With graph partitioning, we end up with 466k auxiliary nodes, and with PPR batching we have 416k auxiliary nodes, across all batches. The number of auxiliary nodes determines the size of the batched feature matrix and thus the memory size. The number of edges per batch also impacts memory usage, but the feature matrix is usually the dominant factor. The edges do, however, have a major impact on GNN computation time. Overall, we should expect a memory usage of around 416/233 = 178% with PPR batching, and 466/233 = 200% with graph partitioning, compared to the normal dataset. All in all, LBMB's memory usage thus depends on three aspects: 1) How large is the training/validation set compared to the full graph? 2) How many auxiliary nodes per primary node are we using? 3) How well are the batches overlapping?
> > > >
> > > > So why are we not seeing a memory overhead in the measured usage for Reddit in Table 6? Because LBMB requires no memory for sampling. Constructing and prefetching batches requires a surprisingly large amount of memory, especially if you aim to saturate main memory or PCIe bandwidth (~30GB/s). The memory used by sampling is required in addition to the normal dataset, while LBMB can delete the dataset from memory after preprocessing.
> > >
> > > **Disk-based datasets:**
> > > Datasets in practice are usually disk-based, since they are too large to recide in any single machine's memory. As you correctly point out, in this case IO time is the critical factor limiting runtime [1]. This is actually a major part of our method's motivation: If the batch recides in a continuous chunk of memory, loading it is much faster -- _especially_ from disk (HDD or SSD), where sequential read speed is >30x faster than random 4k read speed.
> > >
> > > **No luck in determinism:**
> > > Deep learning and ICLR are both full of empirical results that have not yet been properly explained by theory, but are based on evidence. The effectiveness of stochastic gradients for deep learning is actually one great example of this, which is still not fully understood. Our results show that LBMB works best in most cases, _and_ is never far behind the best result. There are no failure cases, as opposed to e.g. GraphSAINT-RW with same method inference. And this makes sense -- after all, our batches are deterministic, there is no luck involved. Especially for inference it seems much _more_ reliable to choose the nodes that impact the output the most, instead of a random set of nodes. This is exactly what the influence score quantifies, which our auxiliary node selection process is based on (see Section 3.2). A random inference method can produce randomly different predictions for a given node (e.g. for a person's cancer treatment decision), which seems like quite the opposite of a reliable and explainable method.

---

> ### Author Response · Authors · 2021-11-21
> **Similar memory usage, strong results for GraphSAGE, and why preprocessing time is secondary (part 1)**
>
> We are happy to hear that you find the question we are tackling important and our idea interesting! We have substantially improved the paper based on your feedback. We think that the revised version and our response should address all of your concerns:
>
> - **Preprocessing vs. training:** The main point to make here is that **preprocessing only has to be done once per batching setting and the result can be saved to disk**. Choosing the batching setting for LBMB was very simple in our experience, since our method effectively only has one hyperparameter: The number of auxiliary nodes per primary node. The rest is either irrelevant or determined by GPU memory. A default of 16 can be used for most practical applications. We finetuned this hyperparameter per dataset and chose one of {8, 16, 32, 64} -- and used the same for all models. So we can fix the LBMB hyperparameters, and run all model development using the same preprocessed dataset. Finding and tuning an ML model architecture and its hyperparameters usually takes hundreds of training runs, usually with multiple seeds. This is orders of magnitude more, and preprocessing only needs to be redone if a new batch size is used -- and even then we can reuse the old PPR values or just merge preprocessed batches. Just considering the 10 seeds we used (not accounting for GNN tuning), for LBMB (PPR batching) **preprocessing only took 1.3% of the training time for GCN and 0.25% for GAT** on ogbn-arxiv. This shows that preprocessing is clearly faster than batch computation during training. Even more compute is spent on inference, which our method excels at -- e.g. according to AWS, 90% of ML infrastructure cost is due to inference [1]. So perhaps a more complete statement would be this: Preprocessing is only performed O(1) times per dataset, training is performed O(100) times more often than preprocessing (using different models and seeds), and ~10 times more compute is spent on inference than on training [1]. Moreover, we have further optimized the preprocessing code and PPR approximation, achieving a 34% reduced preprocessing time for PPR batching and even improving LBMB's accuracy by 0.4 percentage points on average. In particular, we have optimized the batch merging step of distance-based partitioning, and switched from a power iteration-based PPR approximation to a more accurate push-flow algorithm.
> - **Memory cost:** We have **added a table with main memory usage** during training to the paper, after carefully checking that all memory is freed as early as possible. In some cases, LBMB uses more main memory than previous methods due to overlapping batches (e.g. on ogbn-products). However, it can also reduce memory requirements because it ignores irrelevant parts of the graph (e.g. on Reddit). Note that the memory usage of LBMB can be controlled via the number of auxiliary nodes. Overall, LBMB uses a similar amount of memory as previous methods, between a 55% increase and a 39% reduction compared to neighbor sampling. Note that we can control LBMB's memory usage by changing the number of auxiliary nodes per primary node. Main memory usage during training (in GiB):
>
> ||ogbn-arxiv|||ogbn-products|||Reddit|||
> |-|:-:|:-:|:-:|:-:|:-:|:-:|:-:|:-:|:-:|
> ||GCN|GAT|GraphSAGE|GCN|GAT|GraphSAGE|GCN|GAT|GraphSAGE|
> |Neighbor sampling|3.0|3.6|3.1|8.7|7.9|8.5|7.4|7.5|7.1|
> |LADIES|3.0|-|-|6.0|-|-|4.8|-|-|
> |GraphSAINT-RW|3.5|3.6|3.5|9.6|9.6|9.6|8.4|8.5|8.4|
> |Cluster-GCN|3.5|3.4|3.5|7.8|6.0|7.3|6.1|4.2|6.5|
> |LBMB, graph part.|3.5|3.6|3.5|7.9|7.0|7.8|6.3|4.9|6.3|
> |LBMB, PPR batching|3.8|3.8|4.2|13.0|12.3|13.2|4.5|5.3|5.1|

---

### Official Review · Reviewer_myta · 2021-11-02

**Correctness:** 4
**Technical Novelty And Significance:** 3
**Empirical Novelty And Significance:** 2
**Recommendation:** 6
**Confidence:** 4

**Main Review:**

Strengths of the paper

1. The paper proposes a novel approach to deriving minibatches for training graph neural networks.
2. The experiments indicate that the proposed technique achieves a significant reduction in training time and dramatically improves the inference time. The experiments appear to have been executed with care.
3. The ablation study and analysis of training set size provide useful insight into the behaviour of the algorithm.

Weaknesses of the paper
1. The envisioned learning pipeline could be better articulated to make the trade-offs clearer in terms of time required for pre-processing and training time (number of epochs, time per epoch).  Related to this are concerns about how hyperparameter tuning factors into an assessment of the practical training time.
2. The procedure is novel, but Cluster-GCN already proposed the partitioning of the graph as part of the sampling process, so the contribution represents
3. The proposed technique is heuristic, with existing techniques being selected and combined. The derivation of the overall algorithm is thus not based on a principled procedure.

4. The experiments report results for only one choice of batch size. The batch size has a major impact on the performance of the sampling algorithms (both in terms of accuracy and convergence time), so it is important to explore a range of choices. The procedure for selecting the hyperparameters for the baselines is not entirely clear

5. The experiments are only conducted for GAT and GCN. While these may be considered to be representative of many GNNs, they are no longer state-of-the-art. It is not entirely clear that the results will carry over to more recent methods that achieve considerably better performance for the addressed learning tasks.

Recommendation and major reasons for the recommendation

I have recommended a “marginally below the acceptance threshold”.

I think the core idea is clever and the authors have exhibited considerable skill in constructing the overall algorithm. On the other hand, there are several concerns that prevent me from recommending that the paper be accepted.

(1) Novelty & heuristic nature: While I do acknowledge the novelty of the technique, the approach is strongly related to Cluster-GCN, and although the proposed method does represent a clear improvement over that approach, this does limit the novelty of the contribution. The technique is heuristic in nature, and is not developed based on a principled foundation. There is no theoretical characterization of the proposed method.  The results are reported for specific choices of local clustering and graph partitioning, so it’s not entirely clear whether the performance is strongly dependent on these selected methods (having said this, the paper does explore two different types of partitioning, and both yield similar performance, so this points in the direction of the technique performing satisfactorily for any reasonable graph partitioning strategy).

(2) Application in practice and nature of the trade-off: It would be very helpful if the paper better articulated how the method would be applied in practice. In addition to the depicted training time, there is a hidden cost of determining hyperparameters that involves a grid search. The proposed method appears to introduce more hyperparameters than some of the other approaches. Moreover, the selected hyperparameters change from dataset to dataset, suggesting that the hyperparameter selection procedure needs to be performed for every dataset (although perhaps Table 6 hints that this is not the case). It then becomes questionable how valuable it is to reduce the training time for a dataset, since it would form only a small portion of the overall training overhead.

The paper states in Section 3.3 that “a machine learning model is trained only once”, but later makes the argument that the pre-processing overhead involved in constructing the partitions is a one-time overhead compared to the training cost. If the model is only trained once, then is it fair to exclude the pre-processing time from the plots in Figure 1?

(3) Experimental procedure: The authors report that a grid search was conducted to find suitable parameters such that there was a similar memory usage, similar number of iterations per epoch, and maximum validation accuracy. Since there seem to be competing criteria and the ranges of the hyperparameter tuning process are not provided, it is difficult to understand exactly how the hyperparameters were selected.

The performance when training on large graphs tends to be quite sensitive to batch size. It would be much better if results could be included for multiple batch sizes. At the least, it should be clear which range of batch sizes was selected for the hyperparameter tuning process.

Questions

(1) GAT and GCN, while representative of many graph models, are not the state-of-the-art for many learning tasks. Is it clear that the results provided in the paper are likely to generalize to other GNN models? If so, why?

(2) If possible, please provide a clarification of the hyperparameter tuning process, particularly for the baselines.

(3) Please clarify the intended use of the approach (how would it be incorporated in a machine learning pipeline in a practical application). How long does hyperparameter tuning take relative to training time? If it takes as long or longer, is this an important consideration, or is it reasonable to focus entirely on training time? Is there an example scenario where training would be repeated multiple times without a requirement for re-tuning the hyperparameters. Is it clear that the pre-processing time should be excluded when comparing two methods?


**Summary Of The Paper:**

The paper addresses the task of training graph neural networks on large graphs. The authors propose a strategy to extract mini-batches that are locally connected. The technique involves dividing nodes into primary and auxiliary nodes, where the primary nodes are those for which an prediction is formed during the batch (usually training nodes). The auxiliary nodes are used for computation of the predictions. The authors propose two procedures for selecting the primary nodes in a minibatch. One uses distance-based partitioning (a greedy procedure building on personalized page rank as a distance measure). The other applies graph partitioning (METIS). After the primary nodes have been partitioned, auxiliary nodes are selected using local clustering techniques (a form of topic-sensitive PageRank is employed). The authors observe that this construction of mini-batches is also useful in the inference process. The paper proposes methods to mitigate the effect of the introduction of correlation in the gradients that arises due to the proximity-based partitioning. There are also strategies to prevent the optimizer from processing consecutive batches that are very similar.

The paper provides experimental results for GAT and GCN for four large graph datasets, with a comparison to four baselines.  The results provide evidence that the proposed strategy leads to a significant reduction in training time and also has a major benefit during inference. The authors the impact of training set size and conduct an ablation study to verify that all proposed aspects of the proposed method yield performance improvement.


**Summary Of The Review:**

** After modifications and discussion, I have increased the score to "marginally above the acceptance threshold".

The recommendation of the review is “marginally below the acceptance threshold”.

The paper is considered to propose sufficiently novel techniques, although aspects of the approach are similar to prior work. The impact of the paper may be somewhat limited because of the heuristic nature of the proposed methodology.

The paper introduces an increased pre-processing overhead in order to achieve a speed-up in training time. Although the pre-processing time is reported in tables, it is excluded in the figures that compare the overall train times. There appears to be a need for considerable hyperparameter tuning. This process, conducted via grid search, might take considerably longer than the actual reported training.

Although the experiments appear to have been conducted thoroughly, results are reported for only one set of hyperparameters. The selection process of these for the baselines is not particularly clear.

---

> ### Author Response · Authors · 2021-11-21
> **Another GNN, batch size results, hyperparameter tuning, and why preprocessing time is secondary (part 2)**
>
> Other points:
> 1. **Relationship to Cluster-GCN [W2, C1]:** We clearly communicate the relationship to Cluster-GCN in the paper and directly compare to it in our experiments. While Cluster-GCN is indeed related to LBMB, we approach the problem from a very different perspective, and achieve significantly better accuracy, at comparable or lower computational cost. Note that Cluster-GCN ignores the dependence of primary nodes on auxiliary nodes. It thus fails for primary nodes that are close to the cluster boundary, since they require auxiliary nodes outside the cluster.
> 2. **Evidence and  approach [W3, C1]:** We agree that our contribution is largely empirical in nature. However, it is based on general considerations of computer architectures and theoretical insights on the properties of GNNs (e.g. influence scores). Furthermore, we embrace the empirical nature of our paper by presenting experiments that go far beyond the scope presented in previous works: 10 experimental setups, 5 methods, 10 repetitions per method. We only use large datasets and use a single, consistent pipeline that equally optimizes our method and all baselines. We present full convergence curves, inference measurements, timings of all steps, memory usage, and ablation studies.
> 3. **Batch sizes [W4, C3]:** We have **added an experiment** on how the accuracy depends on batch size to our paper, see Fig. 10. LBMB is much less sensitive to this choice than previous methods (e.g. [2]). LBMB can thus even be used in extremely constrained settings with small batches of 100 primary nodes per batch. Above 1000 primary nodes per batch the accuracy varies by at most 0.4 percentage points. We furthermore present inference results using different batch sizes for all methods (at constant GPU memory) in Fig. 4, which concisely presents the accuracy vs. computation trade-offs.
> 4. **Number of hyperparameters [C2]:** As discussed above, our method effectively only has one hyperparameter: The number of auxiliary nodes per primary node. Previous methods often have more hyperparameters, even if we ignore one that determines the batch size. For example, GraphSAINT-RW has three (number of seed nodes, number of random walks per seed node, length of random walks), and neighbor sampling has one per layer (number of samples), i.e. three or often more.
> 5. **Preprocessing vs. training vs. inference [C2]:** In Section 3.3 we highlighted that once a model is found, it is trained only once before being put into production to serve billions of queries. This should stress the large difference in importance between training and inference: E.g. according to AWS, 90% of ML infrastructure cost is due to inference [1]. Perhaps a more complete statement would be this: Preprocessing is only performed O(1) times per dataset, training is performed O(100) times more often than preprocessing (using different models), and ~10 times more compute is spent on inference than on training [1].
>
> [1] https://d1.awsstatic.com/events/reinvent/2019/REPEAT_1_Deliver_high_performance_ML_inference_with_AWS_Inferentia_CMP324-R1.pdf
>
> [2] Dong et al., Global Neighbor Sampling for Mixed CPU-GPU Training on Giant Graphs, KDD'21

---

> > ### Comment · Reviewer_myta · 2021-11-25
> > **After Authors' Response**
> >
> > Thank you for the response. I read the other reviews and your responses with interest, and have closely followed the discussion with one of the other reviewers. I see merits to both sides of the discussion, in that the proposed, deterministic sampling approach is not well understood, but, on the other hand, there can be value in presenting experimental results that demonstrate outperformance.
> >
> > I have increased my score based on the modifications and the discussion.
> >
> > (1) I appreciate the addition of the GraphSAGE model. I would argue that not all advances in GNNs over the past three years have amounted to tricks. Scaling the more complicated algorithms, which do exhibit better performance on smaller datasets, to the larger datasets such as those on the OGB leaderboard is a challenge, perhaps because clever scaling and sampling approaches, such as the one you propose, have not been applied. In this sense, I am not sure that the OGB leaderboard should be viewed as an accurate depiction of the state-of-the-art in GNN methods (although it is perhaps a fair reflection of the state-of-the-art for larger datasets).
> >
> > (2) The improved discussion of the hyperparameter tuning is helpful, although the statement “We performed a grid search on ogbn-arxiv, ogbn-products…” does not clarify the ranges over which the grid search was conducted, nor exactly which variables were included in the grid.
> >
> > (3) The discussion in the responses to reviews has helped clarify a number of issues that were unclear to me about the problem setting and the intended application scenario of the model. I still think it would help if the exact task and the goal were more clearly specified in the introduction to the paper. For example, based on the first two paragraphs, the improvement in inference time is seemingly not a primary goal, but more an inadvertent and beneficial outcome. But then in the discussion with reviewers, there is an argument that inference will be performance thousands of times (“continually for days and for weeks”). With this claim, it is not clear what the setting is – is the graph remaining the same, but the node features changing? Is the graph changing? Or is it a new graph entirely? Is there incremental training? If so, how does this factor into the proposed approach?
> >
> > While I very much like the approach in the paper, I do find it challenging to understand from the paper the exact goal and what the overall improvement is when I factor in hyperparameter tuning, preprocessing, training and inference.

---

> > > ### Author Response · Authors · 2021-11-28
> > > **GNN inference tasks**
> > >
> > > Thank you for reconsidering your score based on the update and discussion! To prevent any loose ends we would like to briefly address the new questions in your reply.
> > >
> > > Please note that we did not mean the term "tricks" in any derogative way. We fully appreciate the progress made in the last three years. We merely wanted to highlight the expected orthogonality between many model improvements and our batching method.
> > >
> > > Investigating the whole breadth of GNN inference tasks and performance would easily fill a new paper. Still, to answer your questions regarding different inference settings with LBMB:
> > > 1. Only node features change: This simply requires re-running the GNN, which would be very efficient to do with LBMB.
> > > 2. Changing graph: LBMB can efficiently handle changing graphs since PPR scores can be updated in parallel, locally, and incrementally [1, 2]. Only large changes to the graph or domain shifts would require repartitioning the batches.
> > > 3. New graph: This requires running LBMB preprocessing on the new graph, which does incur some cost. However, we would expect this to be rather rare, since datasets are usually evolving over time, and not generated in one step. For example, all of the datasets used in our work are in reality evolving over time (citation networks, co-purchase graphs, co-posting graphs).
> > > 4. Incremental training: Continuous learning for GNNs is a largely unexplored topic, so we cannot comment on all aspects of this. From LBMB's perspective, this is essentially the same as a changing graph (point 2).
> > >
> > > We hope that our paper and the discussion in this forum have shown the reviewer why hyperparameter tuning is comparable to previous methods, why preprocessing time is secondary compared to training and inference time, and that our method yields advantages for both training and inference. We are happy to see that these advantages convinced the reviewer to recommend acceptance.
> > >
> > > [1] Bahmani, Chowdhury, Goel. Fast incremental and personalized pagerank. VLDB 2011
> > >
> > > [2] Guo, Li, Sha, Tan. Parallel personalized pagerank on dynamic graphs. VLDB 2017

---

> ### Author Response · Authors · 2021-11-21
> **Another GNN, batch size results, hyperparameter tuning, and why preprocessing time is secondary (part 1)**
>
> We are happy to hear that you find our approach novel, the achieved improvements significant or even dramatic, and our analysis insightful! Thank you for highlighting these open questions and weak spots of our paper! We think that our updated version and the following answer should resolve most of them. We refer to the part of the review that each point addresses using the following keys: Qx = question x, Wx = weakness x, Cx = concern x.
>
> Questions:
> 1. **Other GNNs [Q1, W5]:** If we look at the OGB leaderboards we notice that current SOTA models usually use a simple model like GCN or GAT and then apply multiple tricks to it, such as C&S, UniMP, FLAG, end-to-end language model finetuning, bag of tricks (BoT), or self-KD. The base model is still a regular GNN, and as such we consider these improvements orthogonal to our model. Adding a set of tricks would be an arbitrary choice, which is why we decided to stick with the pure underlying models. We do expect our method to work similarly well with these tricks. Additionally, we have now **added experiments for the GraphSAGE model** (as suggested by reviewer AWEr), resulting in a total of 10 experimental setups and 51 separate main experiments (each executed 10 times) in the paper.
> 2. **Hyperparameter tuning [Q2, W4, C3]:** The 3 tuning priorities described in the paper describe different considerations. 1. To keep methods comparable in a realistic setup, we keep the GPU memory usage constant between methods. This can be viewed as a constraint on how we can choose hyperparameters. 2. When there are semantic hyperparameters that do not influence performance (such as the number of steps per epoch in GraphSAINT-RW, which only changes how an epoch is defined), we choose them to be comparable to other methods. 3. We choose all relevant hyperparameters based on validation accuracy. The same process holds for both our method and all baselines. We have clarified this process in the paper.
> 3. **Preprocessing [Q3, W1, C2]:** Tuning LBMB was fairly simple in our experience. Effectively, our method only has one hyperparameter: The number of auxiliary nodes per primary node. A default of 16 can be used for most practical applications. We finetuned this hyperparameter per dataset and chose one of {8, 16, 32, 64} -- and used the same for all models. The number of primary nodes per batch is then determined by GPU memory, and the PPR hyperparameters can just be ignored, as shown in Table 6. After this we can fix the LBMB hyperparameters, and run all model development using the same preprocessed dataset, **no re-tuning required**. Finding and tuning an ML model architecture and its hyperparameters usually takes hundreds of training runs, usually with multiple seeds. This is orders of magnitude more, and preprocessing only needs to be redone if a new batch size -- and even then we can reuse the old PPR values or just merge preprocessed batches. For example, just considering the 10 seeds we used (not accounting for GNN tuning), for LBMB (PPR batching) **preprocessing only took 1.3% of the training time for GCN and 0.25% for GAT** on ogbn-arxiv. This shows that preprocessing is clearly faster than batch computation during training. Moreover, we have further optimized the preprocessing code and PPR approximation, achieving a 34% reduced preprocessing time for PPR batching and even improving LBMB's accuracy by 0.4 percentage points on average. In particular, we have optimized the batch merging step of distance-based partitioning, and switched from a power iteration-based PPR approximation to a more accurate push-flow algorithm.

---

### Official Review · Reviewer_WNRe · 2021-11-03

**Correctness:** 3
**Technical Novelty And Significance:** 2
**Empirical Novelty And Significance:** 3
**Recommendation:** 5
**Confidence:** 4

**Main Review:**

This paper proposes LBMB, a scalable and effective method for extracting mini batches (subgraphs) by primary node partitioning, followed by auxiliary node selection. This method can be applied to both training and inference, which is a strength of this work. Evaluation demonstrates that LBMB spends much less epoch time for training than existing methods, while maintaining comparable accuracy. Inference also shows significant speedups than existing methods (especially than those using the full graph). However, I still have several concerns/questions about this proposal as follows:

* *Methodology* - While the mini-batch precomputation is the major source of performance gains in LBMB, I am wondering how many sub-processes the authors used for runtime sampling (i.e., *num_workers* in PyTorch). I am asking because a runtime sampling method can often be fully pipelined with GNN computation if a sufficient number of sub-processes are used [R1][R2].

* *Space Overhead* - This work leverages space-time tradeoffs in extracting mini-batches via precomputation. That said, I am curious about the *space (memory) overhead* incurred by the proposed mini-batch precomputation.

* *Cost of Preprocessing* - The preprocessing time seems to be substantially higher than other schemes. The authors argue that the preprocessing is one-time cost, but this overhead can be compounded by hyperparameter tuning for LBMB such as the number auxiliary nodes per primiary node, batch size, and so on. Furthermore, the proposed preprocessing approach seems not feasible to am *inductive* setting where the GNN should deal with unseen data.

* *Applicability of Preprocessing* - Can the preprocessing idea be applied to the other mini batching methods for GNNs such as node-wise and layer-wise sampling methods?


[R1] Dong et al., Global Neighbor Sampling for Mixed CPU-GPU Training on Giant Graphs, KDD'21

[R2] Min et al., Large Graph Convolutional Network Training with GPU-Oriented Data Communication Architecture, VLDB'21



**Summary Of The Paper:**

This paper proposes locality-based mini batching (LBMB), a method for extracting batches for GNN training. LBMB aims to reduce the computation cost of GNN training by treating primary (output) nodes and auxiliary nodes separately.  LBMB can be used for both training and inference, and speeds up both tasks substantially while maintaining comparable accuracy.

**Summary Of The Review:**

This paper proposes a scalable and effective method for extracting mini batches (subgraphs), which can be applied to both training and inference of GNNs. However, I still have several major issues/questions about this work, and would ask the authors to address them in their rebuttal.

---

> ### Author Response · Authors · 2021-11-21
> **Similar memory usage and why preprocessing time is secondary**
>
> We are happy to hear that you agree that our method shows significant speedups in both training and inference, at sustained accuracy. Thank you for highlighting these open questions left by our paper! As far as we can tell, your criticism mostly stems from gaps in our paper, which we address in our revised version. Specifically:
> - **Methodology:** In the paper we use one parallel sub-process for sampling and overlap data loading with processing on GPU. We didn't observe any speed ups when using multiple workers. The reason for this is that this problem is **memory-bound**, not compute-bound. As we stress in the paper, the memory accesses caused by random sampling are highly sub-optimal. Using multiple workers can not solve this. We added a sentence to our paper to clarify this:
>   > To minimize data loading and memory access overhead we always prefetch the next batch in parallel. We found that using more than one worker for data loading does not improve runtime, since loading is memory-bound, not compute-bound.
>
>   Direct main memory access by the GPU as in [R2] or caching on GPU as in [R1] can improve the CPU-to-GPU bottleneck and better saturate its bandwidth. Still, the achieved speed-ups demonstrated in [R1, R2] are lower than the speed-ups achieved by our method of circumventing the random access altogether.
> - **Space overhead:** We have **added a table with main memory usage** during training to the paper, after carefully checking that all memory is freed as early as possible. In some cases, LBMB uses more main memory than previous methods due to overlapping batches (e.g. on ogbn-products). However, it can also reduce memory requirements because it ignores irrelevant parts of the graph (e.g. on Reddit). Overall, LBMB uses a similar amount of memory as previous methods, between a 55% increase and a 39% reduction compared to neighbor sampling. Note that you can control LBMB's memory usage by changing the number of auxiliary nodes per primary node. Main memory usage during training (in GiB):
>
> ||ogbn-arxiv|||ogbn-products|||Reddit|||
> |-|:-:|:-:|:-:|:-:|:-:|:-:|:-:|:-:|:-:|
> ||GCN|GAT|GraphSAGE|GCN|GAT|GraphSAGE|GCN|GAT|GraphSAGE|
> |Neighbor sampling|3.0|3.6|3.1|8.7|7.9|8.5|7.4|7.5|7.1|
> |LADIES|3.0|-|-|6.0|-|-|4.8|-|-|
> |GraphSAINT-RW|3.5|3.6|3.5|9.6|9.6|9.6|8.4|8.5|8.4|
> |Cluster-GCN|3.5|3.4|3.5|7.8|6.0|7.3|6.1|4.2|6.5|
> |LBMB, graph part.|3.5|3.6|3.5|7.9|7.0|7.8|6.3|4.9|6.3|
> |LBMB, PPR batching|3.8|3.8|4.2|13.0|12.3|13.2|4.5|5.3|5.1|
>
> - **Cost of preprocessing:** While preprocessing indeed has to be run multiple times for different batching settings, tuning this was very simple in our experience. **Effectively, our method only has one hyperparameter**: The number of auxiliary nodes per primary node. A default of 16 can be used for most practical applications. We finetuned this hyperparameter per dataset and chose one of {8, 16, 32, 64} -- and used the same for all models. The number of primary nodes per batch is then determined by GPU memory, and the PPR hyperparameters can just be ignored, as shown in Table 6. After this we can fix the LBMB hyperparameters, and run all model development using the same preprocessed dataset. Finding and tuning an ML model architecture and its hyperparameters usually takes hundreds of training runs, usually with multiple seeds. This is orders of magnitude more, and preprocessing only needs to be redone if a new batch size -- and even then we can reuse the old PPR values or just merge preprocessed batches. For example, just considering the 10 seeds we used (not accounting for GNN tuning), for LBMB (PPR batching) **preprocessing only took 1.3% of the training time for GCN and 0.25% for GAT** on ogbn-arxiv. This shows that preprocessing is clearly faster than batch computation during training. For inductive inference, we can calculate PPR very efficiently in an incremental fashion since PPR is calculated independently per node. And for inductive training we can still preprocess all data, since it is known a priori. Moreover, we have further optimized the preprocessing code and PPR approximation, achieving a 34% reduced preprocessing time for PPR batching and even improving LBMB's accuracy by 0.4 percentage points on average. In particular, we have optimized the batch merging step of distance-based partitioning, and switched from a power iteration-based PPR approximation to a more accurate push-flow algorithm.
> - **Applicability of Preprocessing:** Indeed, the preprocessing idea can be applied to many different methods! The question then always becomes: Is random sampling still the best choice, or is there something better? This question lies at the heart of our paper. You might now be asking how cached node-wise sampling compares to our choice of batching. This is exactly what we investigate in one of our ablation studies (Section 5 and Figure 7). Fixed random batching converges slower and does not reach the same level of accuracy as LBMB.

---

> ### Author Response · Authors · 2021-11-28
> **A gentle reminder**
>
> We have made significant improvements to the paper based on your helpful feedback and addressed all of your concerns in our response. The discussion phase is ending very soon. We would be very thankful if you would revisit your evaluation.

---

### Author Response · Authors · 2021-11-21
**Overview of response and update**

We would like to thank all reviewers for their very insightful comments! We found that most concerns and questions overlap, so we would like to give an overview of some of the main paper updates and replies.

- **Preprocessing time and hyperparameters:** Since LBMB is rather easy to tune and invariant to many model choices (see Table 7, Figure 10), we only need to re-run it if we change the batch size due to memory requirements. LBMB's preprocessed batches can instead be saved to disk and reloaded. All GNN model development -- which usually sums up to hundreds of model, hyperparameter, and seed combinations -- can be done from this saved, preprocessed dataset. Just considering the 10 seeds we used (not accounting for GNN tuning), for LBMB (PPR batching) **preprocessing only took 1.3% of the training time for GCN and 0.25% for GAT** on ogbn-arxiv. This is why we find training time to be much more critical than preprocessing time. We have also further optimized the preprocessing code and PPR approximation, achieving a 34% reduced preprocessing time for PPR batching and even improving LBMB's accuracy by 0.4 percentage points on average.
- **Added third GNN:** We have added another model, GraphSAGE (suggested by reviewer AWEr) to our experiments, and show results for all 3 datasets and all 6 methods. LBMB works even better with this model than with GCN or GAT.
- **Main memory usage data:** LBMB uses a similar amount of memory as previous methods -- we observe between a 55% increase and a 39% reduction compared to neighbor sampling. The differences are due to (1) overlapping batches, which causes overhead, and (2) LBMB only using the relevant parts of the graph, which reduces memory. We have added a table listing the memory usage for all settings and methods (Table 6).
- **Batch size experiment:** We have added an experiment investigating LBMB's dependence on batch size (Figure 10). LBMB is much less sensitive to this choice than previous methods (e.g. [R1]). LBMB can thus even be used in extremely constrained settings with small batches of 100 primary nodes per batch.

[R1] Dong et al., Global Neighbor Sampling for Mixed CPU-GPU Training on Giant Graphs, KDD'21

---

### Decision · Program_Chairs · 2022-01-20

**Decision:**

Reject

**Comment:**

The paper suggests a non-random strategy for selecting minibatches of nodes for training graph neural networks. The main argument is that consecutive memory accesses are faster than random accesses, and thus they claim a 20x speedup per epoch by precomputing batches at a small cost to accuracy.

There are a number of discussion points. One reviewer finds the results hard to believe because previous work has shown that runtime sampling can be fully pipelined. The authors agree but say their speedups are still better, which isn’t a fully satisfying response, and it calls into question the quality of the baseline implementation. Another concern is about the effect of deterministic minibatches. The authors argue that the empirical results speak for themselves, while the reviewer worries about robustness. There also are some concerns about methodology around hyperparameters and special-casing of preprocessing for one dataset, though those appear mostly resolved.

On the whole, this is a borderline paper that lands just on the side of rejection. I’d encourage the authors to more thoroughly address the questions about quality of the baseline implementation and the reviewer’s concern about robustness of deterministic minibatches, and then resubmit to the next conference.